# ALIGNMENT-ENHANCED INTEGRATION OF CONNECTIVITY AND SPECTRAL SPARSITY IN DYNAMIC SPARSE TRAINING OF LLM

**Wenjing Wu[1,2], Yingtao Zhang[1,2]\*, Jialin Zhao[1,2], Carlo Vittorio Cannistraci[1,2,3]\***

[1] Center for Complex Network Intelligence (CCNI), Tsinghua University, China[†]
[2] Department of Computer Science and Technology, Tsinghua University, China
[3] Department of Biomedical Engineering, Tsinghua University, China

## ABSTRACT

With the rapid development of large language models (LLMs), identifying efficient strategies for training such large-scale systems has become increasingly critical. Although LLMs have achieved remarkable success across diverse applications, the necessity of maintaining full dense matrices during pre-training has been questioned, giving rise to parameter-efficient sparse pre-training methods which retains parameter-efficiency in both training and inference. These methods can be further divided into connectivity sparse training and spectral sparse training, with dynamic connectivity sparse training and low-rank factorization emerging as representative approaches for the two branches. However, a unified framework that effectively combines the strengths of both has yet to be established. In this work, we observe that the *cancellation effect* between the sparse and low-rank branches may limit the expressivity of the model, manifesting as output conflicts when the two components are combined. To address this issue, we first quantify the cancellation effect using the overlap cancellation ratio (OCR) and then propose a novel scheme that integrates dynamic sparse training with low-rank training, introducing a simple yet effective **alignment loss** to mitigate the disagreement between the two branches and promote better collaboration. We validate this scheme by combining a representative dynamic sparse training method, CHTs, with low-rank training, resulting in a new parameter-efficient training approach termed **CHTsL**. The method is evaluated on LLaMA60M and LLaMA130M using the OpenWebText and C4 datasets, where only 10%, 20%, and 30% of the parameters are preserved compared to dense training. Experimental results demonstrate that our proposed scheme effectively alleviates the cancellation effect, especially in the Q and K matrices of the attention layers, and improves training stability and performance compared to the naive combination of sparse and low-rank components. Additionally, the new scheme enables CHTsL to consistently outperform other parameter-efficient sparse training methods under the same parameter budget, achieving performance closest to that of dense training.

## 1 INTRODUCION

Large language models (LLMs) have attracted tremendous attention due to their superior performance across a wide range of tasks. Despite their impressive capabilities, training LLMs from scratch remains extremely memory-intensive and computation-intensive (Samsi et al., 2023), making it challenging to scale such models under reasonable resource constraints. This has motivated extensive research on efficient methods that reduce computational and memory costs while retaining competitive performance. One of the most direct strategies is to reduce the number of parameters. Early studies on pruning and low-rank fine-tuning (Hu et al., 2022; Zhang et al., 2023; Renduchintala et al., 2023; Sheng et al., 2023; Liu et al., 2024; Kopiczko et al., 2023; Dettmers et al., 2023)

---

\*Corresponding author. Correspondence: kalokagathos.agon@gmail.com
[†]Research Center in Tsinghua Laboratory of Brain and Intelligence (THBI), Department of Psychological and Cognitive Sciences.

have shown that even after removing or compressing a large fraction of parameters, models can still preserve much of their original representational capacity. These findings suggest that parameter-efficient model manipulation is feasible, and they naturally motivate the extension from pruning or finetuning to sparse pretraining, where models are trained from scratch under constrained parameter budgets while maintaining competitive performance compared with dense training.

We divides current approaches to sparse pre-training can be broadly into two branches: connectivity sparse training and spectral sparse training, which refers to those methods utilizing low-rank factorization during pretraining.

The former branch focuses on enforcing sparsity in the connectivity of weight matrices, with dynamic connectivity sparse training emerging as a representative technique(Mocanu et al., 2018; Jayakumar et al., 2020; Evci et al., 2020; Yuan et al., 2021; Zhang et al., 2024; 2025). Dynamic connectivity sparse training maintains a sparse connectivity pattern throughout pre-training, dynamically changing the sparse connectivity and updating active weights to approximate the capacity of dense models. Recent works have shown that on multiple tasks, dynamic sparse training can approach or even surpass the performance of dense models with as little as 10% of the trainable parameters, marking a significant step forward in efficient training(Zhang et al., 2024).

The second branch, spectral sparse training (Zhao et al., 2024a), is typically instantiated through low-rank factorization. Since the low-rank factors are updated during training, the spectral representation they induce also evolves accordingly, which makes spectral sparse training inherently dynamic. Initially proposed in the context of LLM fine-tuning (Hu et al., 2022), low-rank methods decompose weight matrices into low-dimensional components, training only the low-rank representations while freezing the full-rank backbone. These approaches drastically reduce the number of trainable parameters and leverage the pre-trained dense model's representational power. More recent attempts have extended low-rank factorization to the pre-training stage (Lialin et al., 2023; Zhao et al., 2024b; Xia et al., 2024; Meng et al., 2024; Zhao et al., 2024a). However, these methods still require the use of full dense matrices during the forward pass, rather than maintaining the spectral sparse structure consistently from training to inference. Overcoming this limitation, successors like CoLA (Liu et al., 2025) preserve the low-rank structure throughout both training and inference, further validating the feasibility of spectral sparse training.

While previous attempt SLTrain (Han et al., 2024) explored combining sparse and low-rank components, the design remains limited in two key aspects. First, the sparse branch in SLTrain is static, serving only as a supplementary term to spectral sparse training rather than leveraging the full potential of dynamic connectivity sparse methods. Second, SLTrain simply performs a pure summation of sparse and low-rank outputs, without any mechanism to promote effective interaction.

In this work, we take a step in this direction. We observe that naive integration of sparse and low-rank branches often suffers from a cancellation effect, where the two components produce conflicting representations that weaken expressivity and hinder convergence. To address this challenge, we propose a new scheme that integrates dynamic connectivity sparse training with low-rank training under the guidance of alignment loss, which aligns the two branches and promotes cooperative learning. Specifically, we instantiate our framework by combining the advanced dynamic sparse training method CHTs (Zhang et al., 2025) with low-rank factorization, resulting in a new parameter-efficient pre-training approach, CHTsL. Extensive experiments on LLaMA-60M and LLaMA-130M (Touvron et al., 2023a;b) with OpenWebText and C4 show that CHTsL consistently outperforms state-of-the-art parameter-efficient sparse training baselines under the same parameter budget. Notably, with only 10%, 20%, or 30% of parameters preserved relative to dense training, CHTsL achieves performance closest to dense models, which would benefit by retaining efficiency in training, inference, and storage.

Our contributions can be summarized as follows:

**First integration of connectivity sparse and spectral sparse in dynamic sparse training.** We make the first attempt to genuinely integrate connectivity sparse and spectral sparse in dynamic sparse training, with dynamic connectivity and dynamic low-rank representaion. Unlike prior work such as SLTrain, where static connectivity sparsity merely served as a supplement to spectral sparsity, our approach fully leverages the complementary strengths of both paradigms.

**Alignment-enhanced unified scheme.** We identify the cancellation effect as a key obstacle in combining sparse and low-rank branches, where conflicting representations weaken model expressivity. To address this, we introduce the overlapping cancellation ratio (OCR) as a quantitative measure, and propose a unified integration scheme that emphasizes interaction and cooperation rather than naive branch summation. By incorporating an alignment loss, our framework explicitly mitigates conflicts, enhances collaboration, and alleviates the observed cancellation phenomenon in attention Q and K matrices.

**Instantiation with CHTsL and empirical superiority.** We instantiate the framework by combining advanced CHTs with low-rank factorization, yielding the proposed method CHTsL. Extensive experiments across different datasets, models, and parameter budgets demonstrate that CHTsL achieves consistently strong performance, ranking first among all parameter-efficient methods with the same parameter scale, and approaching dense model performance with significantly fewer parameters.

## 2 RELATED WORK

The rapid growth of large language models (LLMs) has stimulated extensive research into improving efficiency in pre-training. Among various directions, *parameter-efficient* approaches have emerged as particularly promising, aiming at training models with limited number of parameters without significantly sacrificing performance. Broadly, parameter-efficient methods in the context of pre-training can be divided into two branches: **connectivity sparse training**, which reduces parameters by enforcing sparse connectivity patterns, and **spectral sparse training**, which constrains weight matrices into low-rank subspaces. Dynamic connectivity sparse training and low-rank factorization are the representative approaches for these two paradigms.

### 2.1 DYNAMIC CONNECTIVITY SPARSE TRAINING

Connectivity sparsity originates from the classical line of pruning (LeCun et al., 1989; Han et al., 2015; Molchanov et al., 2016), where removing parameters from dense models was shown to preserve much of the model's performance. Inspired by this, researchers began to explore whether sparsity could be maintained *throughout training*, rather than applied only as a post-hoc compression. Among these efforts, methods that promote sparse training through dynamic adjustment of connectivity have gained increasing attention, as they often outperform static sparse training approaches that prune connections solely at initialization (Prabhu et al., 2018; Lee et al., 2018; Dao et al., 2022; Stewart et al., 2023). The pioneering work Sparse Evolutionary Training (SET) (Mocanu et al., 2018) removes links while introducing random rewiring of sparse connections during training to maintain model plasticity. RigL (Evci et al., 2020) further dynamically regrows connections based on gradient for more effective exploration, though it requires computing gradients of the full weight matrix during the backward pass. MEST (Yuan et al., 2021) improves upon this by leveraging both weight and gradient information. CHT (Zhang et al., 2024) and its successor CHTs (Zhang et al., 2025) enhance dynamic sparse training using the Cannistraci-Hebbian theory (Muscoloni et al., 2022) from network science, inspired by brain connectomes, achieving state-of-the-art performance on multiple tasks. Collectively, these studies demonstrate that dynamic sparse training can attain competitive or even superior performance compared to dense training, while using only 10% or fewer of the parameters (Zhang et al., 2025).

### 2.2 LOW-RANK FOR SPECTRAL SPARSE TRAINING

Complementary to connectivity sparsity, spectral sparse training leverages low-rank factorization to reduce the dimensionality of weight matrices. This idea was first popularized in the fine-tuning setting, where LoRA (Hu et al., 2022) adapts pretrained models by learning only low-rank updates rather than full weight matrices. Subsequent works (Hu et al., 2022; Zhang et al., 2023; Renduchintala et al., 2023; Sheng et al., 2023; Kopiczko et al., 2023; Dettmers et al., 2023; Liu et al., 2024) further demonstrate the effectiveness of low-rank fine-tuning and inspire the exploration of training from scratch with low-rank factorization. ReLoRA (Lialin et al., 2023) introduces reparameterization to improve training efficiency and stability, while GaLore (Zhao et al., 2024b) reduces memory usage by applying low-rank projections in the gradient space during training. However, a common limitation of these approaches is that the full dense weight matrix is still required during the forward pass, providing parameter efficiency only during training but not during inference. In contrast,

CoLA (Liu et al., 2025) explicitly maintains the low-rank representation throughout both training and inference, enabling reduced storage and runtime costs.

As a side note, while we previously discussed the relevance of pruning mainly in the context of connectivity-based sparse training, in contrast to spectral low-rank training, structured pruning can also be viewed as implicitly inducing a low-rank structure in the resulting model. This is because structured-pruned models remove entire channels or filters, which correspond to removing rows or columns in the unfolded weight matrices, thereby potentially reducing their effective rank. Representative structured-pruning-aware works include channel pruning via LASSO (He et al., 2017), network slimming (Liu et al., 2017), and more recently Only-Train-Once (OTO) (Chen et al., 2021), which explicitly consider structural constraints during training to improve efficiency for subsequent pruning. In this study, we adopt CoLA (Liu et al., 2025) as the baseline under the same restriction of parameter efficiency in both forward and backward passes.

### 2.3 HYBRID ATTEMPT

Beyond individual paradigms, researchers have also begun to explore combining connectivity and spectral sparsity. SLTrain (Han et al., 2024) represents one of the earliest attempts in this direction. It augments low-rank factorization with a sparse branch, but its design exhibits several limitations. Specifically, the sparse component is *static* rather than dynamic, serving merely as a supplementary term to spectral sparsity instead of leveraging genuine connectivity sparse training. Moreover, SLTrain integrates the two branches via a simple summation, without introducing any collaborative mechanism to exploit their potential synergy. As a result, while SLTrain marks an important step toward hybrid parameter-efficient pre-training, it remains an immature solution, leaving room for more principled approaches.

## 3 ALIGNMENT-ENHANCED INTEGRATION OF CONNECTIVITY SPARSE AND SPECTRAL SPARSE

In this section, we present a unified approach for combining dynamic sparse training (connectivity sparse) with low-rank factorization (spectral sparse) under extreme sparsity. While each method alone can improve parameter-efficiency and memory-efficiency, their naive combination often leads to conflicting outputs that limit the model's effective capacity. We address this challenge with three key steps: (i) identifying and quantifying the *cancellation effect*, (ii) introducing a training framework that stabilizes low-rank outputs and encourages cooperation between branches, and (iii) instantiating a method, CHTsL, that integrates connectivity sparse and spectral sparse for dynamic sparse training based on this framework.

### 3.1 CANCELLATION EFFECT AND OCR METRIC

When a sparse branch and a low-rank branch are trained together, a common phenomenon emerges: their outputs sometimes point in opposite directions. This **cancellation effect** means that some portion of the signal from one branch can be neutralized by the other, wasting representational power. In other words, even if each branch individually carries meaningful information, their naive sum may not fully reflect that information, effectively underutilizing the model's capacity.

To quantify this, we define the **Overlap Cancellation Ratio (OCR)**:

$$\text{OCR} = \frac{\sum_i \min(|S_i|, |L_i|) \cdot \mathbf{1}\{S_i L_i < 0\}}{\sum_i \min(|S_i|, |L_i|) + \varepsilon}, \tag{1}$$

where $S$ and $L$ are the outputs of the sparse and low-rank branches, respectively. OCR measures the fraction of overlapping signal that is canceled due to opposite directions, with naturally restricted in the range $[0, 1]$. A higher OCR indicates more severe cancellation.

### 3.2 TRAINING FRAMEWORK: ALIGNMENT LOSS AND ACTIVATION ADJUSTMENT

**Alignment Loss for Cooperative Learning.** When training using two distinct components, the sparse and low-rank branches can produce conflicting signals. Intuitively, if one branch pushes a feature in one direction while the other pushes in the opposite direction, the net effect is reduced

expressivity. To address this cancellation effect, we introduce an **alignment loss** that encourages the outputs to move in similar directions:

$$\mathcal{L}_{\text{align}}^{(l)} = \frac{1}{BN}\|S^{(l)} - L^{(l)}\|_F, \ \mathcal{L}_{\text{align}} = \sum_l \mathcal{L}_{\text{align}}^{(l)}, \tag{2}$$

where $B$ is the batch size and $N$ is the number of elements in one sample's output at layer $l$. This loss penalizes discrepancies between the sparse and low-rank outputs, reducing destructive interference and letting each branch focus on complementary aspects of representation. Each layer contributes to the total alignment loss, which is then weighted in the final objective.

**Activation Adjustment for Low-rank Stability.** Low-rank factorization reduces the number of trainable parameters but can sometimes produce unstable outputs, particularly under extreme sparsity. Inspired by CoLA Liu et al. (2025), we apply a mild non-linear activation between the factorized matrices:

$$L^{(l)} = B^{(l)}\sigma(A^{(l)}x), \tag{3}$$

where $\sigma(\cdot)$ is a non-linear function (SiLU (Hendrycks & Gimpel, 2016) in our experiments). Here, the activation primarily serves to stabilize the low-rank outputs, maintaining a reasonable scale and preventing numerical issues during training. Its role is mainly supportive, ensuring the low-rank branch contributes reliably alongside the sparse branch.

**Overall Objective.** Combining these ideas, the output of each $O^{(l)}$ and the total training loss $L^{(l)}$ are respectively:

$$O^{(l)} = S^{(l)} + L^{(l)}, \ \mathcal{L} = \mathcal{L}_{\text{task}} + \lambda\mathcal{L}_{\text{align}} \tag{4}$$

where $\lambda$ balances the contribution of alignment. This objective ensures that the sparse and low-rank branches are jointly optimized, stabilizing low-rank training and encouraging the two branches to cooperate, mitigating cancellation.

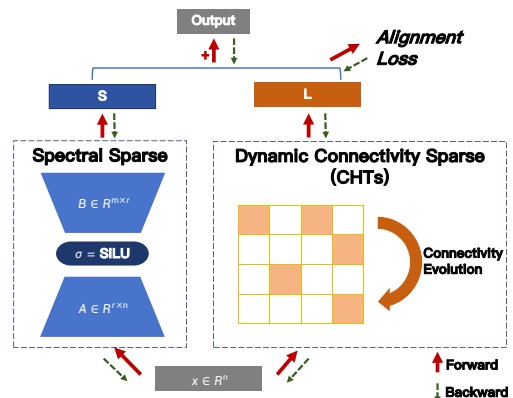

Figure 1: **Workflow of CHTsL.** The figure illustrates CHTsL as an example of alignment-enhanced integration between dynamic connectivity sparse training and spectral sparse training. Specifically, the dynamic connectivity sparse branch adopts the CHTs method.

### 3.3 CHTsL: INSTANTIATING THE FRAMEWORK

Based on this training framework, we propose **CHTsL**, which integrates dynamic connectivity sparse training method CHTs (Zhang et al., 2025) with spectral sparse (low-rank) components. In CHTsL, the sparse branch follows the CHTs update rules, while the low-rank branch incorporates mild activation adjustment, and the alignment loss is applied layer-wise to encourage cooperative outputs. This instantiation demonstrates how our framework naturally combines dynamic connectivity and spectral sparsity, providing a practical approach for training extremely sparse models under a unified scheme. Figure 1 illustrates how CHTsL works.

## 4 EXPERIMENT

### 4.1 MODELS

Experiments are based on Transformer models from the LLaMA family (Touvron et al., 2023a;b), with parameter sizes ranging from 60M to 130M. All models are trained and evaluated on NVIDIA A100 or A800 GPUs.

### 4.2 DATASETS

For training and evaluation, we adopt two widely used large-scale text corpora:

**OpenWebText (Gokaslan & Cohen, 2019)**: A publicly available open-source replication of the WebText dataset used in GPT-2. It is constructed by scraping URLs shared on Reddit with high karma scores, covering a broad range of high-quality web content.

**Colossal Clean Crawled Corpus (C4) (Raffel et al., 2020)**: A large-scale dataset derived from web pages collected through Common Crawl. After extensive cleaning and filtering, it provides high-quality natural language text suitable for large language model pre-training.

## 4.3 BASELINE METHODS

To verify the effectiveness of our method, we compare it against several parameter-efficient training baselines with an equivalent number of trainable parameters. Specifically, we consider dynamic connectivity sparse training methods including SET (Mocanu et al., 2018), RigL (Evci et al., 2020), MEST (Yuan et al., 2021) and CHTs (Zhang et al., 2025); spectral sparse training method CoLA Liu et al. (2025); hybrid method SLTrain (Han et al., 2024). We also report the performance of dense training for comparison.

## 4.4 DEFINITION OF SPARSITY

Since this work integrates connectivity-based sparsity with spectral (low-rank) sparsity, it is necessary to establish a consistent definition of sparsity. For both connectivity sparse and spectral sparse (based on low-rank factorization of a full matrix), we adopt the same definition of sparsity $s$ and corresponding density $d$, representing the fraction of parameters relative to a full-rank dense matrix, which allows fair comparison across methods by reflecting the total number of trainable parameters:

$$s = 1 - \frac{\#\text{params}}{\#\text{params}_{\text{dense}}}, \quad d = 1 - s. \tag{5}$$

For a connectivity sparse method, the original sparsity corresponds to the true sparsity of the network. For a low-rank factorization of dense matrices of size $m \times n$ with rank $r$, the effective density is $(m + n)r/(m \cdot n)$. For a method that integrates both connectivity and spectral sparsity, the total sparsity can be computed as

$$s_{\text{total}} = 1 - d_{\text{connectivity}} - d_{\text{spectral}}. \tag{6}$$

In our experiments, all methods are compared under the same total sparsity to ensure an equivalent number of trainable parameters. For clarity in the Section 5, we report the total sparsity of each method, and we additionally provide the **sparsity-configuration** for the integrated methods, which includes the sparsity $s$ of the connectivity sparse component, the rank $r$ of the low-rank component, and the proportion $\frac{d_{connectivity}}{d_{spectral}}$ of parameters between two branches in Appendix B.

## 4.5 HYPERPARAMETER SETTINGS

Alignment-enhanced training scheme introduces the coefficient $\lambda$ to control the effect of alignment loss. We searched the $\lambda$ in the range [0, 0.1, 0.3 0.5, 0.7, 1] with preliminary experiments. For LLaMA-60M on OpenWebText and LLaMA-130M, the appropriate $\lambda$ is 0.5; For LLaMA-60M on C4, the appropriate $\lambda$ is 0.3.

For methods combining sparse and low-rank training (including SLTrain and CHTsL), the sparsity-configuration mentioned in Section 4.4 need to be considered under the same total parameter budgets. We systematically varied the allocation of parameters between the sparse and low-rank branches in steps corresponding to total sparsity of 5% and the best results across all sparsity-configurations were reported. The step size for rank adjustment in the low-rank branch was calculated based on the model architecture, resulting in approximate step values of 16 for LLaMA-60M and 24 for LLaMA-130M, of which the concrete calculation process can be found in Appendix A.

All the other hyperparameters can be found in Appendix B, which is set to be the same maximally for different methods for a fair comparison.

## 5 RESULT AND DISCUSSION

In this section, we present the experimental results. We first present the effectiveness of the alignment-enhanced training scheme by comparing it with the naive integration of CHTs and low-rank factorization. And then we compare different efficient training methods under the same parameter budget to present that CHTsL consistently improves the performance, realizing the performance most close to dense training with limited parameters.

Table 1: **Comparison between different integration strategies.** The table consists of two parts: **a. The performance of different integration strategies**, reported in terms of validation perplexity (PPL↓). The *Naive* strategy corresponds to a simple sum of CHTs and low-rank factorization. The *Act* strategy applies activation adjustment to the low-rank factorization branch. The *Act+Align* strategy combines activation adjustment with the alignment loss. The coefficient of the alignment loss $\lambda$ is reported in Section 4.5. The sparsity configuration is set such that the sparse branch and the low-rank branch have the same number of trainable parameters($\frac{d_{connectivity}}{d_{spectral}} = 1$). **b. The Wilcoxon signed-rank test p-values**, which indicate whether the differences in performance between strategies are statistically significant.

| Model | Dataset | Total Sparsity | *Naive* | *Act* | *Act+Align* |
|---|---|---|---|---|---|
| LLaMA-60M | OpenWebText | 0.9 | 32.64 | 32.21 | **31.77** |
| | | 0.8 | 33.35 | 29.42 | **29.11** |
| | | 0.7 | 27.89 | 29.94 | **27.40** |
| | C4 | 0.9 | 189.55 | 39.66 | **39.29** |
| | | 0.8 | 36.71 | 36.54 | **36.16** |
| | | 0.7 | 591.42 | 34.55 | **34.33** |
| LLaMA-130M | OpenWebText | 0.9 | 119.35 | 24.45 | **24.07** |
| | | 0.8 | 22.11 | 21.98 | **21.87** |
| | | 0.7 | 21.12 | 20.90 | **20.65** |
| | C4 | 0.9 | 30.77 | 30.30 | **30.03** |
| | | 0.8 | 27.83 | 27.68 | **27.59** |
| | | 0.7 | 920.16 | 26.55 | **26.19** |
| Wilcoxon signed-rank | against *Naive* | | \ | 0.0093 | 0.00049 |
| p-value | against *Act* | | \ | \ | 0.00049 |

## 5.1 Effectiveness of alignment-enhanced integration

**Performance improvement** To verify the effectiveness of the alignment-enhanced training scheme, we compare CHTsL with the naive integration between CHTs and low-rank factorization. In Table 1, we present the result of CHTs plus low-rank with different integration strategy on different models and datasets with different total sparsity, under the constraint that sparse component and low-rank component dominates the same number of parameters ($\frac{d_{connectivity}}{d_{spectral}} = 1$). The results are validated by Wilcoxon signed-rank test for the statistical comparison. It shows that, with $p-value < 0.05$, activation adjust of low-rank improves the training stability and the whole alignment training scheme makes CHTsL significantly better than the naive integration.

**Eased Cancellation Effect.** Figure 2 presents the OCR defined in Equation 1, comparing the cancellation effect between the naive integration and the alignment-enhanced approach for the experiment on LLaMA-60M with the OpenWebText dataset under a total sparsity of 0.9 , with sparsity-configuration $s = 0.95, r = 16, \frac{d_{connectivity}}{d_{spectral}} = 1$ . We observe that incorporating the alignment loss significantly reduces the OCR in the Query and Key layers, with performance substantially surpassing that of the naive integration. A plausible explanation is that Q and K, as the core components of attention, directly determine the attention weights via their dot product, making them highly sensitive to inconsistencies between the dynamic sparse branch and the low-rank branch. Enforcing alignment therefore stabilizes the attention maps and mitigates gradient conflicts, whereas feed-forward or value projections are more tolerant to internal variations due to residual connections. Consequently, this targeted consistency in Q and K enhances the robustness of the attention mechanism, leading to overall performance improvements. More evidence of experiments under other sparsity levels can be found in Appendix C.

## 5.2 CHTsL outperforms other sparse training methods

Table 2 reports the results of CHTsL in comparison with all baseline methods under the same total parameter budget. The results demonstrate that CHTsL consistently outperforms all other methods given the same parameter constraint. This provides strong evidence for the potential of integrating connectivity sparse training with spectral sparse training, achieving performance closest to dense training while preserving only 30% or fewer of the training parameters.

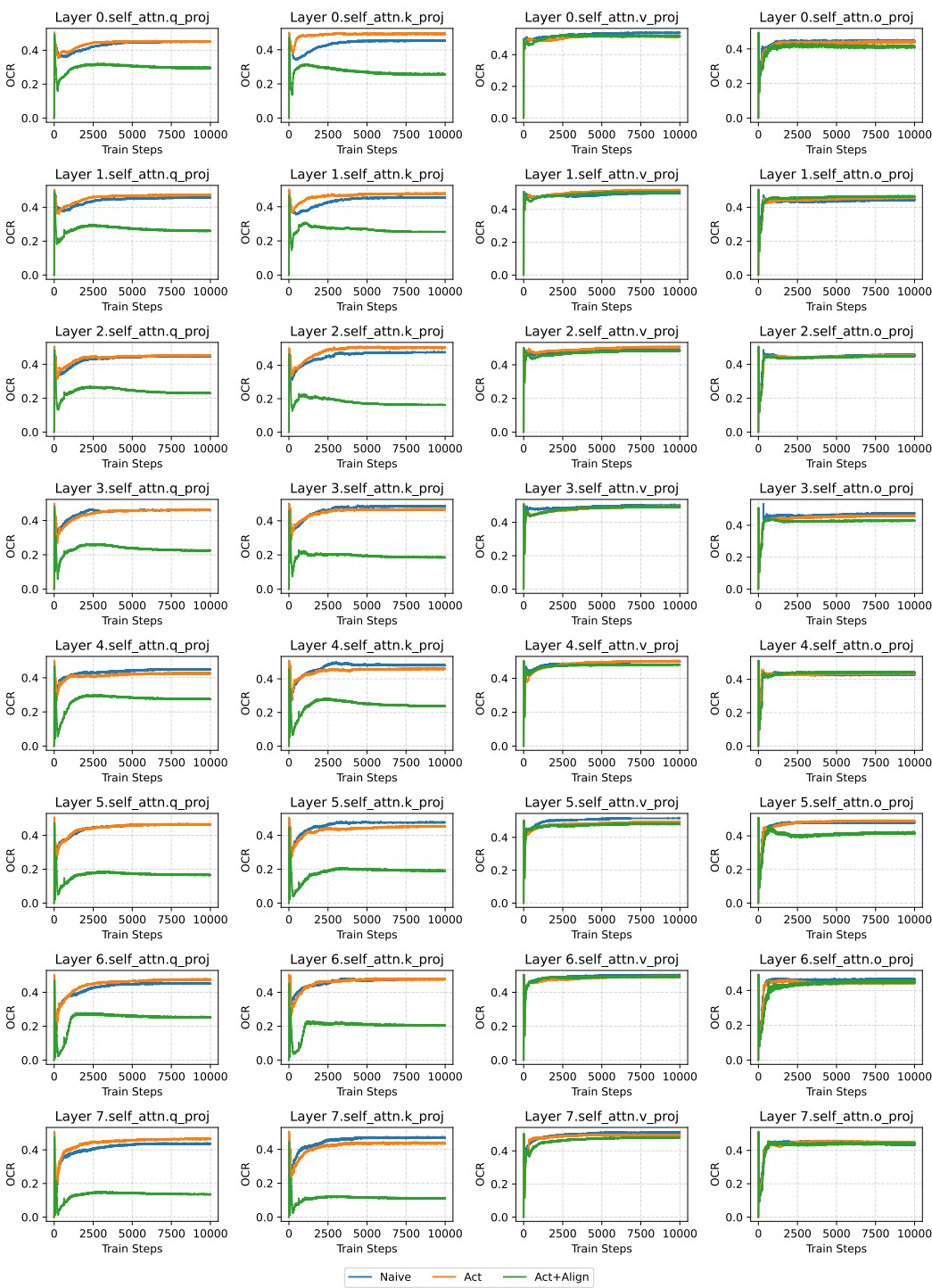

Figure 2: **The layer-wise OCR plot** of LLaMA60M on OpenWebText with a total sparsity of 0.9, with sparsity-configuration $s = 0.95, r = 16, d_{connectivity} : d_{spectral} = 1 : 1$. Each subplot in the figure reports the changes of OCR over training steps. The plot is based on the experiment of the first row of Table 1. For space limit, we report here the self-attention layers in the model, where each column refers to Q, K, V, O respectively.

Table 2: **Validation perplexity of different methods.** Validation perplexity (PPL↓) is reported in this table for different methods on different datasets under the same constraint of total sparsity $s_{total}$. Bold values are the best performance out of all sparse methods.

| Dataset | Method | LLaMA-60M | | | LLaMA-130M | | |
|---|---|---|---|---|---|---|---|
| | | $s_{total}$=0.9 | $s_{total}$=0.8 | $s_{total}$=0.7 | $s_{total}$=0.9 | $s_{total}$=0.8 | $s_{total}$=0.7 |
| OpenWebText | Dense | | 26.56 | | | 19.46 | |
| | SET | 35.26 | 30.69 | 31.77 | 25.70 | 23.20 | 22.03 |
| | RigL | 45.34 | 41.33 | 39.96 | 41.25 | 44.49 | 70.11 |
| | MEST | 33.6 | 29.94 | 28.26 | 25.59 | 22.93 | 21.63 |
| | CHTs | 33.03 | 29.84 | 28.12 | 24.75 | 22.67 | 21.48 |
| | CoLA | 37.58 | 30.87 | 28.53 | 27.07 | 23.24 | 21.61 |
| | SLTrain | 33.90 | 29.83 | 27.86 | 25.33 | 22.81 | 21.25 |
| | CHTsL | **31.77** | **29.11** | **27.40** | **24.07** | **21.87** | **20.65** |
| C4 | Dense | | 33.21 | | | 24.55 | |
| | SET | 42.32 | 37.70 | 35.62 | 32.45 | 29.47 | 27.75 |
| | RigL | 53.39 | 48.59 | 47.34 | 43.57 | 55.82 | 64.93 |
| | MEST | 41.46 | 37.28 | 35.40 | 32.54 | 29.29 | 27.59 |
| | CHTs | 40.62 | 37.55 | 35.23 | 31.00 | 28.69 | 27.46 |
| | CoLA | 46.41 | 38.58 | 35.87 | 33.52 | 29.26 | 27.25 |
| | SLTrain | 41.05 | 37.00 | 34.89 | 31.38 | 28.28 | 26.78 |
| | CHTsL | **39.29** | **35.95** | **34.19** | **30.03** | **27.59** | **26.19** |

Figure 3: **Sensitivity analysis of sparsity configurations under a total sparsity of 0.7.** The sparsity-configuration is defined by the sparsity $s$ in the connectivity-sparse branch and the rank $r$ in the low-rank branch. Each subplot illustrates the variation of validation perplexity (PPL↓) as the rank decreases by step of 5% total sparsity. Outliers with PPL values exceeding the corresponding thresholds are highlighted in red, with their true values explicitly annotated.

### 5.3 SENSITIVITY TEST FOR SPARSITY CONFIGURATIONS

In Figure 3, we illustrate how validation perplexity (PPL↓) varies with different sparsity configurations across models and datasets under a fixed total sparsity of 0.7. On OpenWebText, when the low-rank branch dominates the parameter budget far more than the connectivity-sparse branch (sparsity in the connectivity sparse branch exceeds 0.9), performance collapses. This instability may be attributed to the dataset's relatively limited complexity. Since OpenWebText is more homogeneous, the model becomes more sensitive to imbalanced sparsity allocation. By contrast, on C4, which

contains substantially more diverse and heterogeneous text, a higher proportion of low-rank parameters proves beneficial. A possible explanation is that the increased variety of linguistic patterns likely requires broader adaptations of the entire weight matrix, making low-rank components more effective in capturing such variability.

## 6 CONCLUSION

In this work, we present a novel framework for parameter-efficient pre-training by systematically integrating connectivity sparse training with spectral sparse in dynamic sparse training. We identify the cancellation effect in naive integration as the key challenge, where conflicting representations branches reduce expressivity and hinder convergence. To address this, we introduce the overlapping cancellation ratio to quantify the effect and an alignment loss to encourage cooperative learning. Building on this framework, we instantiate CHTsL by combining the advanced dynamic sparse training method CHTs with low-rank factorization. Extensive experiments on LLaMA-60M and LLaMA-130M with OpenWebText and C4 demonstrate that CHTsL consistently outperforms existing methods under equivalent parameter budgets. Our work is the first to systematically unify dynamic connectivity and spectral sparse training, moving beyond static connectivity sparsity and naive integration; it identifies and mitigates the cancellation effect, fostering effective collaboration between the sparse and low-rank components; and it provides a practical instantiation that validates the benefits of this integration. Overall, this study offers both theoretical insights and practical solutions for efficient sparse pre-training, highlighting the potential of combining complementary sparsity paradigms to maximize model expressivity under constrained resources.

### ACKNOWLEDGMENTS

This work was supported by: The National Natural Science Foundation of China, Research Fund for International Scientists - III (grant number W2531064, to C.V.C). The Zhou Yahui Chair Professorship award of Tsinghua University (to C.V.C). The National High-Level Talent Program of the Ministry of Science and Technology of China (grant number 20241710001, to C.V.C). The authors thank Yuchi Liu and Mo Yang for their administrative support.

### REPRODUCIBILITY STATEMENT

The code for this work is provided in the supplementary material. Detailed hyperparameter settings for each method are presented in Appendix B to facilitate reproducibility.

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

# A    Sparsity Configuration for LLaMA-60M and LLaMA-130M

The sparsity configuration for methods combining a sparse branch with a low-rank branch is defined by two values: $s$, the sparsity of the connectivity-sparse component, and $r$, the rank of the low-rank component.

In our experiments, for each fixed total sparsity, we varied the sparsity-configuration in steps of $\Delta s = 0.05$. That is, whenever the parameter count of one branch was reduced by 5% relative to dense training, the parameter count of the other branch was increased accordingly. Since the sparsity of the connectivity-sparse branch is directly tied to the total sparsity, the main challenge is determining the corresponding rank adjustment in the low-rank branch, which depends on the structure of the LLaMA model.

All linear layers in LLaMA are replaced by our sparse components. Because LLaMA models of different sizes are built from repeated Transformer blocks with identical architecture, it suffices to analyze a single block to establish the relationship between $s$ and $r$. Each block contains seven linear layers, denoted as Q, K, V, O, up, down, and gate. Among them, Q, K, V, and O have weight matrices of size $h \times h$, while up, down, and gate have size $h \times f$, where $h$ is the embedding dimension and $f$ is the feed-forward dimension. Hence, the step size of the rank $r_{\text{step}}$ corresponding to $\Delta s = 0.05$ is determined by:

$$\frac{4(h+h)r_{\text{step}} + 3(h+f)r_{\text{step}}}{4(h \times h) + 3(h \times f)} = \Delta s = 0.05. \tag{7}$$

For LLaMA-60M with $h = 512$ and $f = 1376$, this yields a rank step size of approximately $r_{\text{step}} \approx 16$. For LLaMA-130M with $h = 768$ and $f = 2048$, the corresponding step size is $r_{\text{step}} \approx 24$.

# B    Detailed Hyperparameter Settings for Each Method

For fair comparison, almost all experiments adopt the common hyperparameter settings listed in Table 3, consistent with prior work.

Table 3: **Common hyperparameter settings** for experiments on LLaMA-60M and LLaMA-130M. The settings align with previous research.

| Hyperparameter | LLaMA-60M | LLaMA-130m |
|---|---|---|
| Embedding Dimension | 512 | 768 |
| Feed-forward Dimension | 1376 | 2048 |
| Global Batch Size | 512 | 512 |
| Sequence Length | 256 | 256 |
| Training Steps | 10000 | 20000 |
| Warmup Steps | 1000 | 2000 |
| Learning Rate | 3e-3 | 3e-3 |
| Optimizer | Adam | Adam |
| Layer Number | 8 | 12 |
| Head Number | 8 | 12 |
| Iterative warmup steps | 20 | 20 |
| Update Interval for DST | 100 | 100 |

There are several exceptions, particularly for dense training and CoLA. For dense training, due to the substantially larger number of parameters, a high learning rate leads to model collapse. Therefore, we adopt a learning rate of 1e-3, following the setup in Zhang et al. (2025). For CoLA, we observed strong sensitivity to the choice of optimizer: using Adam causes training collapse (with perplexity exceeding 100). To stabilize training, we use the AdamW optimizer provided in their official code.

Method-specific hyperparameter settings are as follows:

**DST methods (SET, RigL, MEST, CHTs):** We follow the hyperparameter configurations reported in Zhang et al. (2025). Specifically, results for LLaMA-60M on OpenWebText are directly imported from Zhang et al. (2025). For experiments not covered in that study, we set $r = 0.25$ for the BRF initialization of CHTs, as it was reported to yield the highest win rate.

**CoLA:** Apart from the hyperparameters in Table 3, we use the same settings as those provided in the official code release.

**SLTrain:** The coefficient $\alpha$ that controls the contribution of the low-rank branch is set to 32 for LLaMA-60M and 16 for LLaMA-130M, following Han et al. (2024). We also found SLTrain to be highly sensitive to the sparsity-configuration (i.e., the allocation of parameters between branches) under total sparsities of $[0.9, 0.8, 0.7]$. To provide reliable results and fair comparison, we searched configurations with a step size of 0.05 sparsity (corresponding to rank steps of 16 for LLaMA-60M and 24 for LLaMA-130M). The best configurations are summarized in Table 4.

**CHTsL:** We employ CHTs with BRF initialization and set $r = 0$. The alignment loss coefficient $\lambda$ is set to 0.5 for LLaMA-60M on OpenWebText and LLaMA-130M, and 0.3 for LLaMA-60M on C4. The sparsity-configuration is tuned with a step size of 0.05. As shown in Section 5.3, the best configurations consistently converge to 1:1 allocation between the two branches on OpenWebText, and $s = 0.95$ on C4. A full summary of the best configurations is provided in Table 5.

Table 4: **The best sparsity-configuration for SLTrain** under different total sparsity. $s_{total}$ refers to total sparsity, $s$ refers to sparsity in the connectivity sparse branch, $r$ refers to the rank in low-rank branch. The last column reports the proportion of parameters in connectivity sparse branch compared with spectral sparse (low-rank) branch.

| Dataset | Model | Sparsity-Configuration | | | |
|---|---|---|---|---|---|
| | | $s_{total}$ | $s$ | $r$ | $d_{connectivity} : d_{spectral}$ |
| OpenWebText | LLaMA-60M | 0.9 | 0.95 | 16 | 1:1 |
| | | 0.8 | 0.9 | 32 | 1:1 |
| | | 0.7 | 0.85 | 48 | 1:1 |
| | LLaMA-130M | 0.9 | 0.95 | 24 | 1:1 |
| | | 0.8 | 0.85 | 24 | 3:1 |
| | | 0.7 | 0.85 | 72 | 1:1 |
| C4 | LLaMA-60M | 0.9 | 0.95 | 16 | 1:1 |
| | | 0.8 | 0.9 | 32 | 1:1 |
| | | 0.7 | 0.9 | 64 | 1:2 |
| | LLaMA-130M | 0.9 | 0.95 | 24 | 1:1 |
| | | 0.8 | 0.95 | 72 | 1:3 |
| | | 0.7 | 0.85 | 72 | 1:1 |

Table 5: **The best sparsity-configuration for CHTsL** under different total sparsity. $s_{total}$ refers to total sparsity, $s$ refers to sparsity in the connectivity sparse branch, $r$ refers to the rank in low-rank branch. The last column reports the proportion of parameters in connectivity sparse branch compared with spectral sparse (low-rank) branch.

| Dataset | Model | Sparsity-Configuration | | | |
|---|---|---|---|---|---|
| | | $s_{total}$ | $s$ | $r$ | $d_{connectivity} : d_{spectral}$ |
| OpenWebText | LLaMA-60M | 0.9 | 0.95 | 16 | 1:1 |
| | | 0.8 | 0.9 | 32 | 1:1 |
| | | 0.7 | 0.85 | 48 | 1:1 |
| | LLaMA-130M | 0.9 | 0.95 | 24 | 1:1 |
| | | 0.8 | 0.9 | 48 | 1:1 |
| | | 0.7 | 0.85 | 72 | 1:1 |
| C4 | LLaMA-60M | 0.9 | 0.95 | 16 | 1:1 |
| | | 0.8 | 0.95 | 48 | 1:3 |
| | | 0.7 | 0.95 | 80 | 1:5 |
| | LLaMA-130M | 0.9 | 0.95 | 24 | 1:1 |
| | | 0.8 | 0.95 | 72 | 1:3 |
| | | 0.7 | 0.95 | 120 | 1:5 |

## C  EASED CANCELLATION EFFECT UNDER ALIGNMENT-ENHANCED INTEGRATION

In this section, we present the OCR curves of different integration schemes across various total sparsity levels for LLaMA-60M on OpenWebText, as a supplement to Section 5.1. Figures 4 and 5 show the OCR curves under total sparsity levels of 0.8 and 0.7, respectively, where the sparsity configuration is constrained such that the two branches contain the same number of trainable parameters. These results correspond to the second and third rows of Table 1, respectively.

## D  USAGE OF LLM

In this work, Large Language Model (LLM) is primarily used to assist with tasks such as text refinement, summarization, and improving the clarity and readability of the manuscript. The LLM helps streamline writing and editing, ensuring that technical content is clearly and accurately presented.

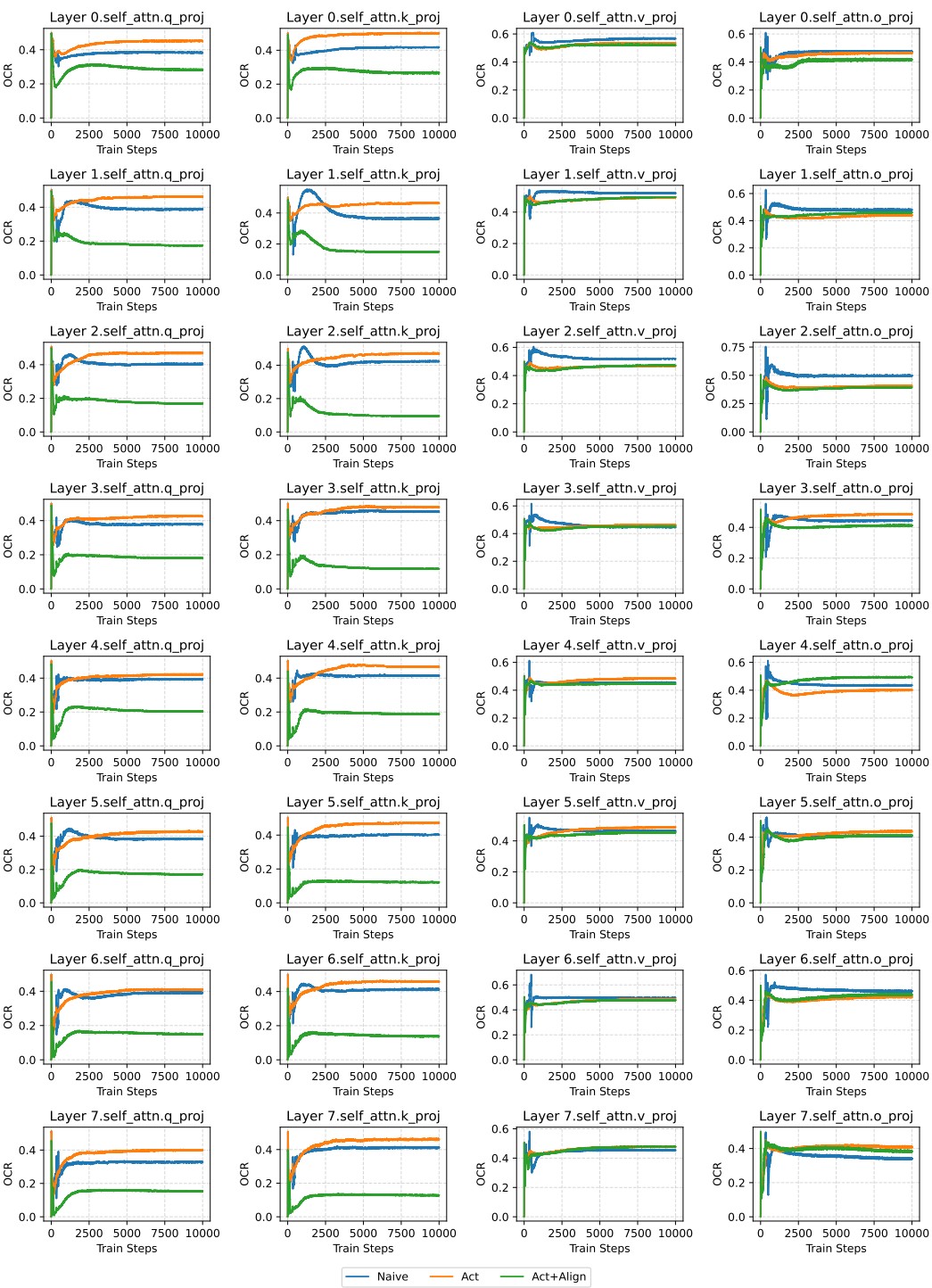

Figure 4: **The layer-wise OCR plot** of LLaMA60M on OpenWebText with a total sparsity of 0.8, with sparsity-configuration $s = 0.9, r = 32, d_{connectivity} : d_{spectral} = 1 : 1$. Each subplot in the figure reports the changes of OCR over training steps. The plot is based on the experiment of the second row of Table 1. For space limit, we report here the self-attention layers in the model, where each column refers to Q, K, V, O respectively.

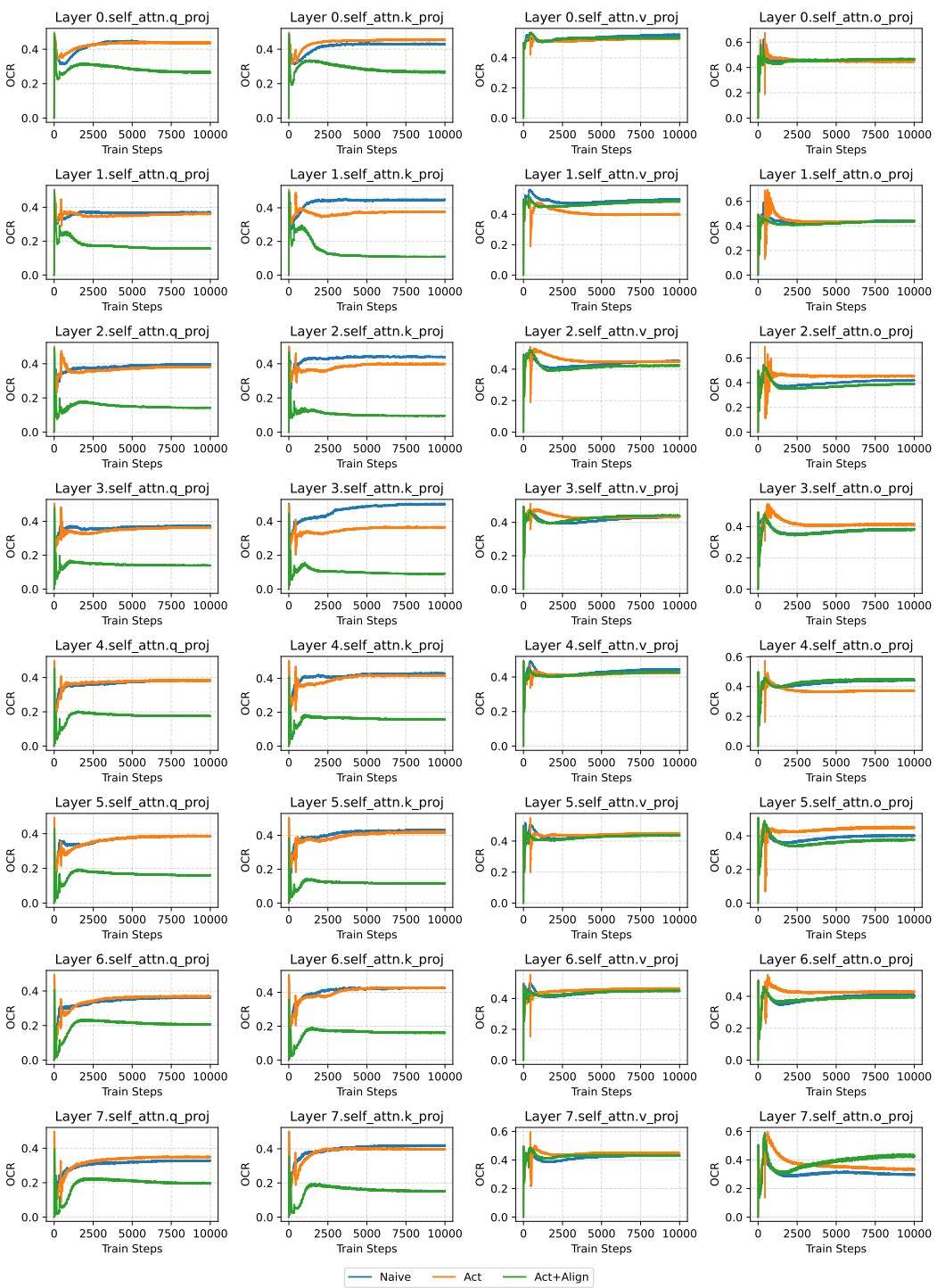

Figure 5: **The layer-wise OCR plot** of LLaMA60M on OpenWebText with a total sparsity of 0.7, with sparsity-configuration $s = 0.85, r = 48, d_{connectivity} : d_{spectral} = 1 : 1$. Each subplot in the figure reports the changes of OCR over training steps. The plot is based on the experiment of the third row of Table 1. For space limit, we report here the self-attention layers in the model, where each column refers to Q, K, V, O respectively.

# E  ALIGNMENT SCHEME ON STATIC SPARSE TRAINING WITH LOW-RANK TRAINING

To further validate the effectiveness of the proposed alignment-enhanced integration scheme, we additionally evaluate it in the "static sparse + low-rank" setting by comparing models trained with and without alignment. Table 6 reports results on LLaMA-130M across multiple datasets and total sparsity levels, under the constraint that the connectivity-sparse and low-rank components occupy the same number of parameters ($\frac{d_{\text{connectivity}}}{d_{\text{spectral}}} = 1$). Statistical significance is confirmed using the Wilcoxon signed-rank test. With $p$-value $< 0.05$, the alignment-enhanced model achieves significantly better performance than the naive integration baseline.

Table 6: **Comparison between different integration strategies for "Static + Low-rank" Combination.** The table consists of two parts: **a. The performance of different integration strategies**, reported in terms of validation perplexity (PPL↓). The *Naive* strategy corresponds to a simple sum of static sparse and low-rank factorization. The *Act* strategy applies activation adjustment to the low-rank factorization branch. The *Act+Align* strategy combines activation adjustment with the alignment loss. The coefficient of the alignment loss $\lambda$ is 0.3. The sparsity configuration is set such that the sparse branch and the low-rank branch have the same number of trainable parameters($\frac{d_{connectivity}}{d_{spectral}} = 1$). **b. The Wilcoxon signed-rank test p-values**, which indicate whether the differences in performance between strategies are statistically significant.

| Model | Dataset | Total Sparsity | *Naive* | *Act* | *Act+Align* |
|---|---|---|---|---|---|
| LLaMA-130M | openwebtext | 0.9 | 31.52 | 25.44 | **25.41** |
| | | 0.8 | 22.44 | 22.37 | **22.36** |
| | | 0.7 | 21.25 | 41.51 | **20.97** |
| | c4 | 0.9 | 31.49 | 31.62 | **31.44** |
| | | 0.8 | 28.21 | 28.35 | **28.12** |
| | | 0.7 | 26.90 | 26.52 | **26.36** |
| Wilcoxon signed-rank | against *Naive* | | \ | 1 | 0.03125 |
| p-value | against *Act* | | \ | \ | 0.03125 |

## F  OCR AND GLOBAL COSINE SIMILARITY

To better understand the cancellation phenomenon between the sparse and low-rank branches, we compare the proposed Overlap Cancellation Ratio (OCR) with commonly used global cosine similarity. While these metrics are related, they capture fundamentally different aspects of cancellation.

### F.1  CONCEPTUAL DIFFERENCE

Global cosine similarity measures only directional alignment between vectors but ignores magnitude, which is crucial for assessing the severity of cancellation. In contrast, OCR explicitly quantifies the fraction of overlapping magnitude that is canceled due to opposite signs.

This distinction can be illustrated by the following examples, both with cosine similarity equal to $0$:

- **Example A (strong cancellation):** $S = [10, -10]$, $L = [10, 10]$. Then $S + L = [20, 0]$, indicating half cancellation. OCR $= 0.5$ ((ignoring the small $\epsilon$ in the denominator).
- **Example B (minimal cancellation):** $S = [100, 0]$, $L = [0, 100]$. Then $S + L = [100, 100]$, indicating almost no cancellation. OCR $= 0$.

These examples demonstrate that while cosine similarity may indicate similar or opposite directions, it does not capture the magnitude of signal lost. OCR complements cosine similarity by explicitly measuring this magnitude-based cancellation.

### F.2  EMPIRICAL OBSERVATION WITH COSINE SIMILARITY

To provide a thorough observation, we include here cosine similarity plots corresponding to the training of LLaMA-60M on OpenWebText under different total sparsities (0.9, 0.8, 0.7) as a supplement. The settings are exactly the same as the reported in Section 5.1 and Section C where OCR plots are reported. Figures 6, 7, and 8 show the evolution of cosine similarity between the outputs of the sparse and low-rank branches during training.

These curves demonstrate that the alignment-enhanced training scheme increases directional alignment between the two branches, with a notably higher cosine similarity observed in the Q and K layers. This observation complements the OCR measure, confirming that the alignment loss not only reduces magnitude-based cancellation but also improves global directional alignment.

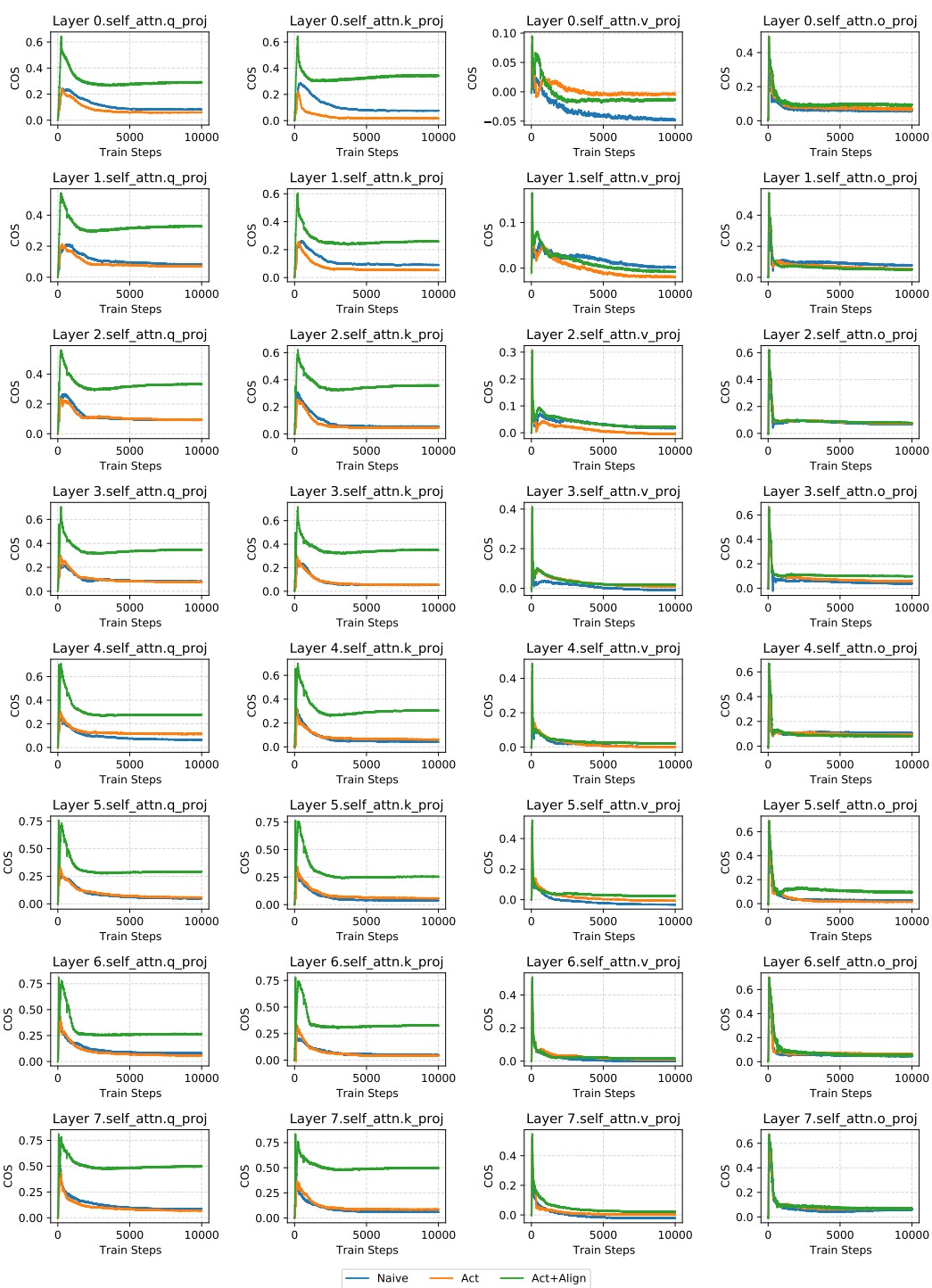

Figure 6: **The layer-wise cosine similarity plot** of LLaMA60M on OpenWebText with a total sparsity of 0.9, with sparsity-configuration $s = 0.95, r = 16, d_{connectivity} : d_{spectral} = 1 : 1$. Each subplot in the figure reports the changes of OCR over training steps. The plot is based on the experiment of the first row of Table 1. For space limit, we report here the self-attention layers in the model, where each column refers to Q, K, V, O respectively.

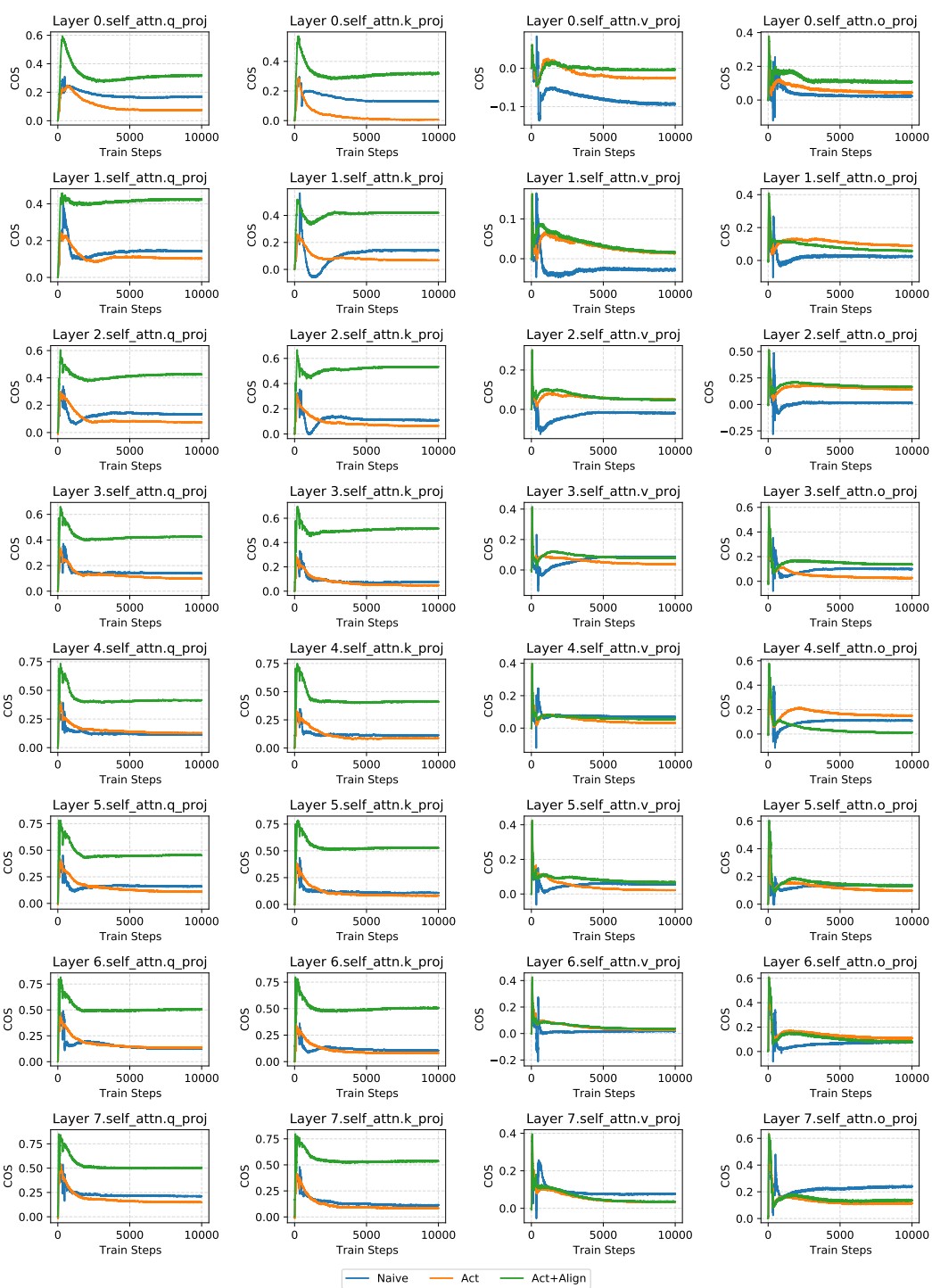

Figure 7: **The layer-wise cosine similarity plot** of LLaMA60M on OpenWebText with a total sparsity of 0.8, with sparsity-configuration $s = 0.9, r = 32, d_{connectivity} : d_{spectral} = 1 : 1$. Each subplot in the figure reports the changes of OCR over training steps. The plot is based on the experiment of the second row of Table 1. For space limit, we report here the self-attention layers in the model, where each column refers to Q, K, V, O respectively.

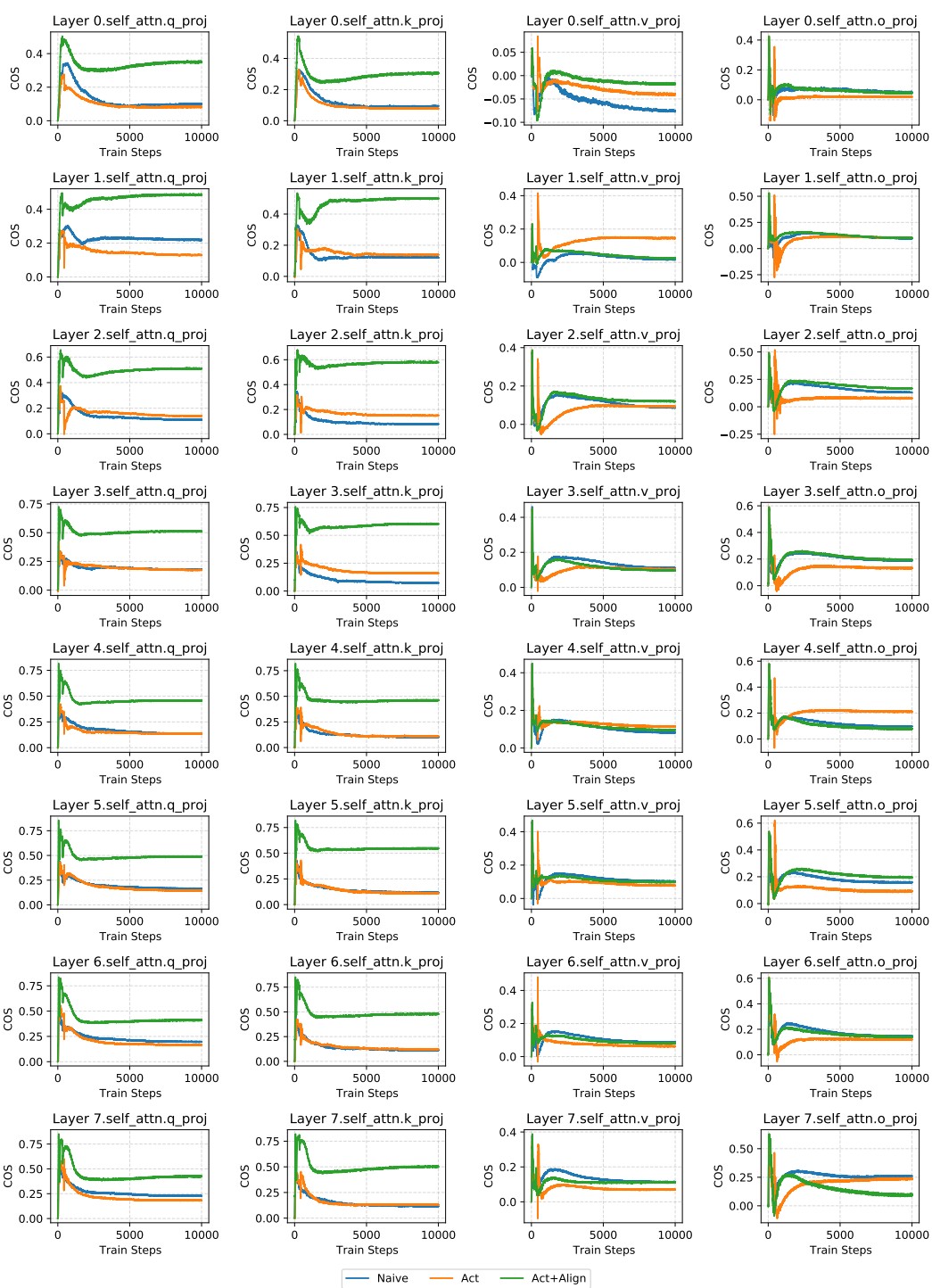

Figure 8: **The layer-wise cosine similarity plot** of LLaMA60M on OpenWebText with a total sparsity of 0.7, with sparsity-configuration $s = 0.85, r = 48, d_{connectivity} : d_{spectral} = 1 : 1$. Each subplot in the figure reports the changes of OCR over training steps. The plot is based on the experiment of the third row of Table 1. For space limit, we report here the self-attention layers in the model, where each column refers to Q, K, V, O respectively.

# G   FURTHER DISCUSSION: APPLYING ALIGNMENT LOSS ONLY TO Q,K LAYERS

According to the OCR plots, when applying the alignment loss to all linear layers, the cancellation effect is primarily mitigated in the Q and K layers rather than uniformly across all layers. This observation motivates a more efficient approach: applying the alignment loss exclusively to the Q and K layers.

To investigate this, we conducted experiments with CHTsL applying alignment loss only to Q and K layers (*Align_qk*) and only except Q/K layers (*Align_others*), compared with those obtained by applying alignment loss to all layers (Align_all), using exactly the same best hyperparameter settings as summarized in Section B. The results, presented in Table 7, demonstrate that restricting the alignment loss to Q and K layers yields similar or even improved performance compared to applying it to all layers. Align Q/K layers significantly outperforms align other linear layers, with the Wilcoxon signed-rank p-value lower than 0.05.

These findings further validate that OCR accurately captures the cancellation effect and highlight that the alignment loss can be implemented more efficiently, achieving comparable or better results with reduced computational cost. Moreover, this observation provides a useful perspective for future work building upon this study, suggesting that targeted alignment may be sufficient to achieve strong performance while saving computation.

Table 7: **Validation perplexity of models based on alignment to different components.** Align_qk refers to CHTsL with alignment only to Q, K layers, while Align_all refers to the original CHTsL with alignment to all linear layers. Validation perplexity (PPL↓) is reported in this table for different methods on different datasets under the same constraint of total sparsity $s_{total}$. Bold values are the best performance.

| Model | Dataset | Total Sparsity | Align_qk | Align_other | Align_all |
|---|---|---|---|---|---|
| LLaMA-60M | OpenWebText | 0.9 | 32.012 | 32.224 | **31.772** |
| | | 0.8 | **29.066** | 29.353 | 29.109 |
| | | 0.7 | **27.279** | 27.802 | 27.400 |
| | C4 | 0.9 | 39.376 | 39.713 | **39.291** |
| | | 0.8 | **35.860** | 36.281 | 35.949 |
| | | 0.7 | 34.194 | 34.437 | **34.191** |
| LLaMA-130M | OpenWebText | 0.9 | 24.251 | 24.400 | **24.071** |
| | | 0.8 | 21.878 | **21.772** | 21.866 |
| | | 0.7 | 20.690 | 26.655 | **20.648** |
| | C4 | 0.9 | 30.135 | 30.454 | **30.034** |
| | | 0.8 | **27.567** | 27.798 | 27.593 |
| | | 0.7 | **26.143** | 38.250 | 26.190 |
| Average score | | | 29.04 | 30.76 | **29.01** |
| Win Rate | | | 0.42 | 0.08 | **0.5** |
| signed-rank p-value | against | Align_qk | \ | 0.00098 | 0.62207 |
| | against | Align_other | \ | \ | 0.00098 |

# H    RESULTS ON MODELS WITH LARGER SIZE

We conducted experiments on the larger LLaMA-350M and LLaMA-1B on OpenWebText to further evaluate the effectiveness of CHTsL at scale. Due to limitations in time and computational resources, we selected SLTrain, CHTs and CoLA as the most competitive sparse training baselines. Also, we provide the performance of dense model for reference.

The common hyperparameter settings for all methods are listed in Table 8. All hybrid methods, including CHTsL and SLTrain, use a sparsity configuration of $d_{connectivity} : d_{spectral} = 1 : 1$.

For CHTsL, the coefficient $\lambda$ which controls the contribution of alignment loss is set to be 0.5.

For SLTrain, the coefficient $\alpha$ controlling the contribution of the low-rank branch is set to 16 for LLaMA-350M and 8 for LLaMA-1B, following Han et al. (2024).

For CHTs under sparsity level 0.7 on LLaMA-1B, we directly imported the result reported in Zhang et al. (2025), with learning-rate 3e-3.

For dense model, since the number of trainable parameters is different from the sparse training methods, we applied smaller learning rate following previous literature Zhang et al. (2025). For LLaMA-350M, the adopted learning rate is 1e-3, while for LLaMA-1B, we directly imported the result reported in Zhang et al. (2025) with learning rate 4e-4.

The results are reported in Table 9. They show that CHTsL consistently achieves lower perplexity across different total sparsity levels compared to the baselines on LLaMA-350M. On LLaMA-1B, CHTsL outperforms all other methods at the 0.9 sparsity level, where the benefits of our approach become particularly pronounced, while at the 0.7 sparsity level it remains competitive, though slightly below CoLA. Overall, CHTsL demonstrates robust performance on large-scale models, underscoring its scalability and its strong advantage especially under higher sparsity regimes.

Table 8: **Common hyperparameter settings** for experiments on LLaMA-350M and LLaMA-1B. The settings align with previous research.

| Hyperparameter | LLaMA-350M | LLaMA-1B |
|---|---|---|
| Embedding Dimension | 1024 | 2468 |
| Feed-forward Dimension | 2736 | 5461 |
| Global Batch Size | 512 | 512 |
| Sequence Length | 256 | 256 |
| Training Steps | 60000 | 100000 |
| Warmup Steps | 6000 | 10000 |
| Learning Rate | 3e-3 | 1e-3 |
| Optimizer | Adam | Adam |
| Layer Number | 24 | 24 |
| Head Number | 16 | 32 |
| Iterative warmup steps | 20 | 20 |
| Update Interval for DST | 100 | 100 |

Table 9: **Validation perplexity of different methods on LLaMA-350M.** Validation perplexity (PPL↓) is reported in this table for different methods on different datasets under the same constraint of total sparsity $s_{total}$. Bold values are the best performance out of all sparse methods.

| Dataset | Method | LLaMA-350M | | | LLaMA-1B | |
|---|---|---|---|---|---|---|
| | | s_total=0.9 | s_total=0.8 | s_total=0.7 | s_total=0.9 | s_total=0.7 |
| OpenWebText | Dense | | 14.90 | | | 14.62 |
| | CHTs | 19.69 | 17.82 | 17.88 | 17.35 | 14.53 |
| | CoLA | 20.92 | 17.60 | 16.13 | 16.03 | **13.08** |
| | SLTrain | 18.99 | 16.88 | 15.98 | 16.00 | 14.58 |
| | CHTsL | **18.40** | **16.54** | **15.86** | **15.16** | 13.31 |

# I    ABLATION TEST FOR ACTIVATION FUNCTION

As an ablation study, we conducted experiments on the low-rank branch using different activation functions, including ReLU and GeLU, in comparison with SiLU. The hyperparameter settings were kept the same as those reported in Section B.

The results, presented in Table 10, show that CHTsL with SiLU activation in the low-rank branch outperforms the alternatives in most cases.

Table 10: **Validation perplexity of CHTsL based on different activation function.** SiLU is the default one used in the main text. Validation perplexity (PPL↓) is reported in this table for different methods on different datasets under the same constraint of total sparsity $s_{total}$. Bold values are the best performance.

| Model | Dataset | Total Sparsity | ReLU | GeLU | SiLU |
|---|---|---|---|---|---|
| LLaMA-60M | OpenWebText | 0.9 | 32.431 | 32.081 | **31.772** |
| | | 0.8 | 29.607 | 29.280 | **29.109** |
| | | 0.7 | 28.125 | 35.625 | **27.400** |
| | C4 | 0.9 | 39.930 | 39.364 | **39.291** |
| | | 0.8 | 36.871 | 36.045 | **35.949** |
| | | 0.7 | 35.251 | **34.157** | 34.191 |
| LLaMA-130M | OpenWebText | 0.9 | 24.530 | **24.049** | 24.071 |
| | | 0.8 | 22.219 | 21.999 | **21.866** |
| | | 0.7 | 20.999 | 20.789 | **20.648** |
| | C4 | 0.9 | 30.639 | 30.171 | **30.034** |
| | | 0.8 | 28.252 | 27.762 | **27.593** |
| | | 0.7 | 27.044 | 26.192 | **26.190** |

## J    ZERO-SHOT EVALUATION

To further evaluate the generality of CHTsL, we assessed the trained models on downstream datasets from GLUE and SuperGLUE. We compared CHTsL with the strongest sparse training baselines, SLTrain and CHTs, as well as with the dense model. All sparse training methods were evaluated under a total sparsity of 0.9, with corresponding hyperparameter settings listed in Section B. Experiments were conducted using the `lm-eval` package, and accuracy (Acc) is reported.

The results, presented in Table 11, show that CHTsL achieves the highest win rate among the sparse training baselines and also outperforms the dense model. These findings further demonstrate the generality and effectiveness of CHTsL across downstream tasks.

Table 11: **Zero-shot results on downstream tasks.** CHTsL, SLTrain, and CHTs are evaluated under a total sparsity of 0.9. Results are reported in terms of accuracy (Acc), with the best-performing value in each row highlighted in bold. **Note that** if two or more methods achieve the same accuracy, all corresponding values are bolded and counted toward the win rate.

| Model | Pretrain | Downstream | CHTsL | SLTrain | CHTs | Dense |
|---|---|---|---|---|---|---|
| LLaMA-60M | OpenWebText | CoLA | 0.5292 | 0.6894 | **0.6913** | **0.6913** |
| | | Copa | 0.5300 | **0.5700** | **0.5700** | 0.5200 |
| | | Hellaswag | 0.2653 | 0.2649 | **0.2663** | 0.2619 |
| | | MNLI | 0.3278 | **0.3310** | 0.3290 | 0.3282 |
| | | MRPC | 0.3235 | 0.3995 | **0.6838** | 0.6789 |
| | | QNLI | **0.4955** | 0.4935 | 0.4944 | 0.4946 |
| | | QQP | **0.4126** | 0.3688 | 0.3682 | 0.3682 |
| | | RTE | **0.5235** | 0.5126 | 0.4838 | 0.5018 |
| | | SST-2 | **0.5482** | 0.4908 | 0.4908 | 0.4920 |
| | C4 | CoLA | **0.6913** | 0.6903 | 0.4276 | 0.6846 |
| | | Copa | 0.4800 | 0.5200 | **0.5400** | 0.4500 |
| | | Hellaswag | 0.2666 | 0.2644 | **0.2714** | 0.2656 |
| | | MNLI | 0.3255 | **0.3340** | 0.3291 | 0.3281 |
| | | MRPC | 0.6544 | 0.6495 | 0.6324 | **0.6740** |
| | | QNLI | **0.4944** | 0.4915 | 0.4926 | 0.4939 |
| | | QQP | 0.3682 | 0.3727 | 0.3692 | **0.3730** |
| | | RTE | **0.5487** | 0.4874 | 0.5271 | 0.5162 |
| | | SST-2 | 0.4908 | **0.4931** | 0.4908 | 0.4908 |
| LLaMA-130M | OpenWebText | CoLA | 0.6606 | 0.6568 | **0.6913** | **0.6913** |
| | | Copa | 0.5700 | 0.5600 | 0.5600 | **0.6000** |
| | | Hellaswag | 0.2687 | 0.2656 | 0.2678 | **0.2699** |
| | | MNLI | 0.3254 | **0.3320** | 0.3275 | 0.3272 |
| | | MRPC | 0.5662 | 0.3775 | **0.6814** | 0.6740 |
| | | QNLI | **0.4961** | 0.4950 | 0.4941 | 0.4946 |
| | | QQP | **0.4015** | 0.3762 | 0.3682 | 0.3717 |
| | | RTE | **0.5018** | 0.4477 | 0.4585 | 0.4765 |
| | | SST-2 | **0.5161** | 0.4920 | 0.4908 | 0.4908 |
| | C4 | CoLA | 0.6002 | 0.6443 | **0.6903** | **0.6903** |
| | | Copa | 0.4700 | 0.4900 | 0.5300 | **0.5700** |
| | | Hellaswag | 0.2713 | 0.2694 | 0.2701 | **0.2770** |
| | | MNLI | **0.3282** | 0.3274 | 0.3276 | 0.3274 |
| | | MRPC | **0.5686** | 0.3211 | 0.4412 | 0.5074 |
| | | QNLI | **0.5041** | 0.5003 | 0.4946 | 0.4946 |
| | | QQP | 0.3720 | **0.4326** | 0.3683 | 0.3682 |
| | | RTE | **0.5199** | 0.5162 | 0.4946 | 0.5018 |
| | | SST-2 | 0.4908 | **0.4920** | 0.4908 | 0.4908 |
| | Win Rate | | **0.4167** | 0.1944 | 0.25 | 0.25 |

## K  INFERENCE MEMORY AND THROUGHPUT

In this section, we report the inference memory usage and throughput for CHTsL, SLTrain, and the dense baseline. For CHTsL and SLTrain, the sparsity configuration was set to $d_{\text{connectivity}}$ : $d_{\text{spectral}} = 1 : 1$, under a total sparsity 0.9. Each model was run for 5000 inference steps with dummy inputs of batch size 128 and sequence length 256. We record the maximum memory usage using `torch.cuda.max_memory_allocated` (in GB) and measure the average throughput (Tokens/sec). Experiments were conducted on a single NVIDIA A100-80GB GPU.

The inference memory and throughput of CHTsL are theoretically identical to those of SLTrain, as the two differ only in the training procedure while sharing the same inference-stage architecture. Both methods gain efficiency from:

- sparse matrix multiplication in the connectivity-sparse branch
- low-rank multiplication in the spectral-sparse branch.

For accurate inference benchmarking, we evaluated the trained CHTsL checkpoints using the SLTrain C++ codebase. For CHTsL, two minimal modifications were made: (1) adding the activation function required by the spectral-sparse branch, and (2) correcting the computation order in the low-rank branch, which in the original SLTrain implementation computed $B@A$ and then $B@A@X$ and thus introduced redundant operations.

Table 12 shows that CHTsL achieves both **lowest memory usage** and **highest throughput** out of three models. The advantage in throughput arises from the corrected low-rank computation order, as mentioned above in the second modification. This demonstrates that CHTsL already provides practical efficiency benefits while holding even greater theoretical potential.

Finally, we emphasize that neither CHTsL nor SLTrain can fully realize their theoretical speed-ups due to current software and hardware limitations. PyTorch does not provide efficient kernels for unstructured sparsity, and modern GPUs offer minimal acceleration for unstructured sparse operations. Thus, all unstructured sparse methods are currently operating below their theoretical limits.

Overall, CHTsL outperforms the dense baseline in both memory and speed, and its efficiency advantage is likely to increase further as frameworks and hardware improve support for unstructured sparsity.

Table 12: **Inference memory and throughput of different methods.** For each model, inference was conducted for 5000 steps, with maximum memory and average throughput reported. Experiments are conducted on 1 x NVIDIA A100-80GB, with dummy input of batch size 128 and sequence length 256.

| Model | Method | Memory (GB) | Throughput (Tokens/s) |
|---|---|---|---|
| | Dense | 2.606 | 773111 |
| LLaMA-60M | SLTrain | **2.573** | 749985 |
| | CHTsL | **2.573** | **786004** |
| | Dense | 3.392 | 343310 |
| LLaMA-130M | SLTrain | **3.278** | 337149 |
| | CHTsL | **3.278** | **386483** |

## L    GRADIENT LEVEL CANCELLATION

To further examine the cancellation effect at the gradient level, we analyze the input gradients of the two branches (i.e., the gradients of x with respect to Wx and Bf(Ax)). Experiments were conducted on LLaMA-60M with OpenWebText under a total sparsity level of 0.9 and a sparsity configuration of $d_{connectivity} : d_{spectral} = 1 : 1$, corresponding to the first row of Table 1.

We plot the OCR curves between the input gradients of the two branches. Figure 9 compares the curves of the Naive combination with those of the alignment-enhanced integration. The results show that, at the gradient level, certain layers exhibit a reduced cancellation effect (e.g., the 5th Q layer). However, the phenomenon is less pronounced than at the output level, which is expected since the alignment loss directly acts on the outputs rather than on the gradients.

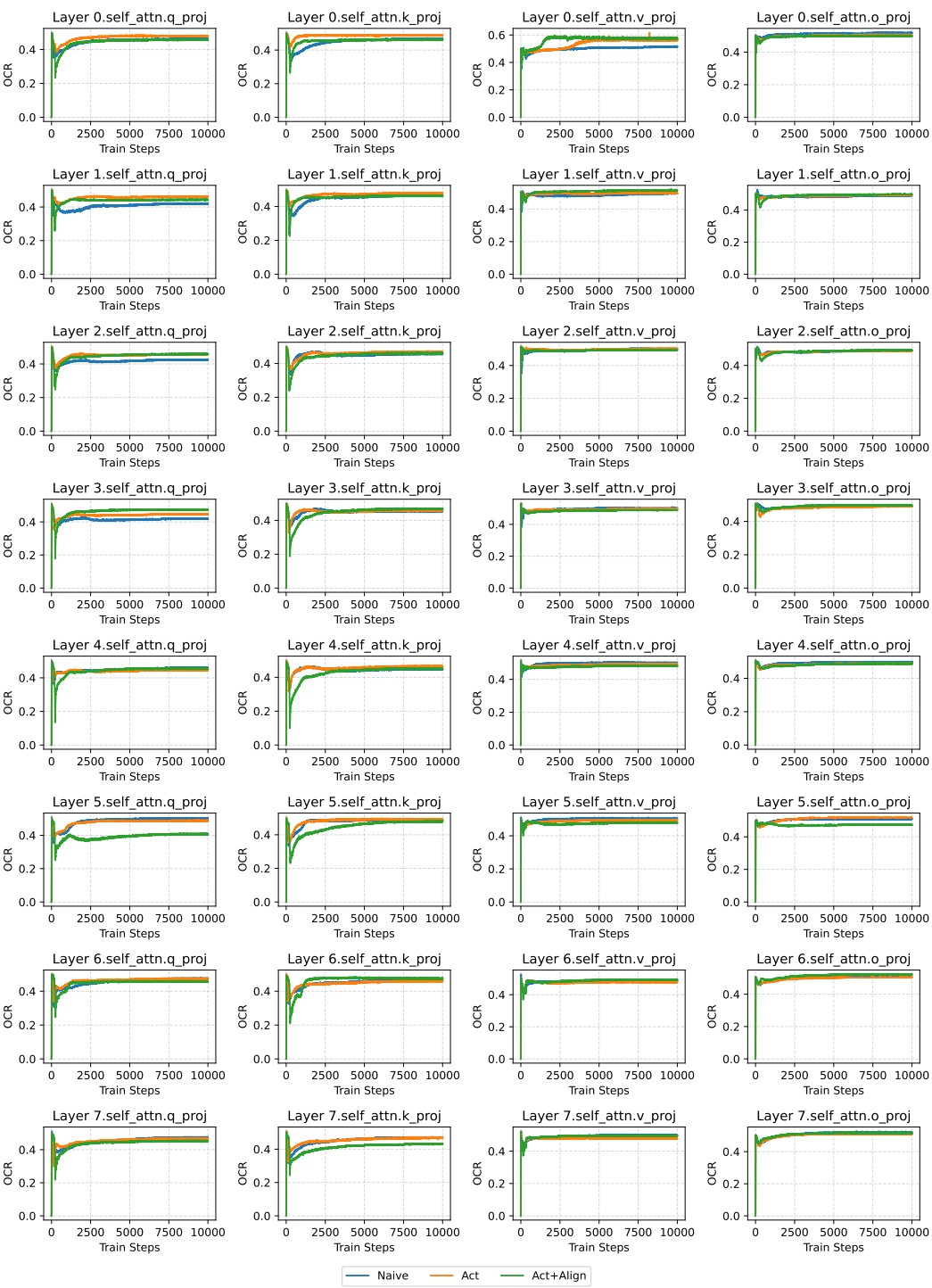

Figure 9: **The layer-wise gradient-level OCR plot** of LLaMA60M on OpenWebText with a total sparsity of 0.9, with sparsity-configuration $s = 0.95, r = 16, d_{connectivity} : d_{spectral} = 1 : 1$. Each subplot in the figure reports the changes of OCR over training steps. The plot is based on the experiment of the first row of Table 1. For space limit, we report here the self-attention layers in the model, where each column refers to Q, K, V, O respectively.

# M  ALIGNMENT ON DIFFERENT COMBINATION OF CONNECTIVITY SPARSITY AND SPECTRAL SPARSITY

In this section, we present results demonstrating how alignment works when combining different connectivity-based sparse training methods with low-rank training.

We conducted experiments using connectivity-based sparse training methods including static sparse training, SET, and CHTs, with different initialization strategies such as random and BSW in the work of  Zhang et al. (2025).

We compare naive integration with alignment-enhanced integration on LLaMA-130M using the OpenWebText dataset, under a total sparsity level 0.9 with a sparsity configuration of $d_{\text{connectivity}}$ : $d_{\text{spectral}} = 1 : 1$. The coefficient $\lambda$ was chosen from 0.3 and 0.5.

Results in Table 13 show that the alignment-enhanced training scheme consistently improves performance compared with naive integration. Moreover, for dynamic connectivity-based sparse training method CHTs, BSW initialization outperforms random initialization, which is consistent with previous literature and further validates the reliability of our results. These findings further confirms the generality of the alignment training scheme on combining connectivity sparse training and spectral sparse training.

Table 13: **Validation perplexity under different combination of connectivity sparsity** and spectral sparsity. *Naive* refers to simple integration of connectivity sparsity. *Act+Align* refers to activation and alignment-enhanced integration.  Bold value refers to the better performance considering different integration strategy.

| DST | Initialization | Naive | Act+Align |
|---|---|---|---|
| Static | random | 22.44 | **22.36** |
| | BSW | 380.88 | **21.88** |
| SET | random | 356.08 | **22.54** |
| | BSW | 22.55 | **22.22** |
| CHTs | random | 22.46 | **22.20** |
| | BSW | 22.11 | **21.87** |

# N   EASED CANCELLATION EFFECT ON LLaMA-350M

In this section, we present the OCR curves of LLaMA-350M with and without the alignment training scheme. Due to time constraints, we compare only the naive integration approach, which simply sums the outputs of the two branches, with the proposed alignment scheme, which leverages both the activation in the low-rank branch and an explicit alignment loss.

All experiments are conducted under an overall sparsity of 0.9, using the sparsity configuration $s = 0.95, r = 32, d_{connectivity} : d_{spectral} = 1 : 1$. The validation perplexity is reported in Table 14, where the performance of the naive integration collapses. The OCR curves in Figure 10 during training show that, under the alignment training scheme, the OCR value decreases significantly, especially in the Q, K layers, as also observed in smaller models. In contrast, the OCR of the naive integration is highly unstable, which stems from the training collapse.

Table 14: **Validation perplexity on LLaMA-350M** under different integration strategy of CHTs and low-rank trainin. *Naive* refers to simple integration. *Act+Align* refers to activation and alignment-enhanced integration. Bold value refers to the better performance considering different integration strategy.

| Model | Dataset | Sparsity | Naive | Act+Align |
|---|---|---|---|---|
| LLaMA-350M | OpenWebText | 0.9 | 604.48 | **18.40** |

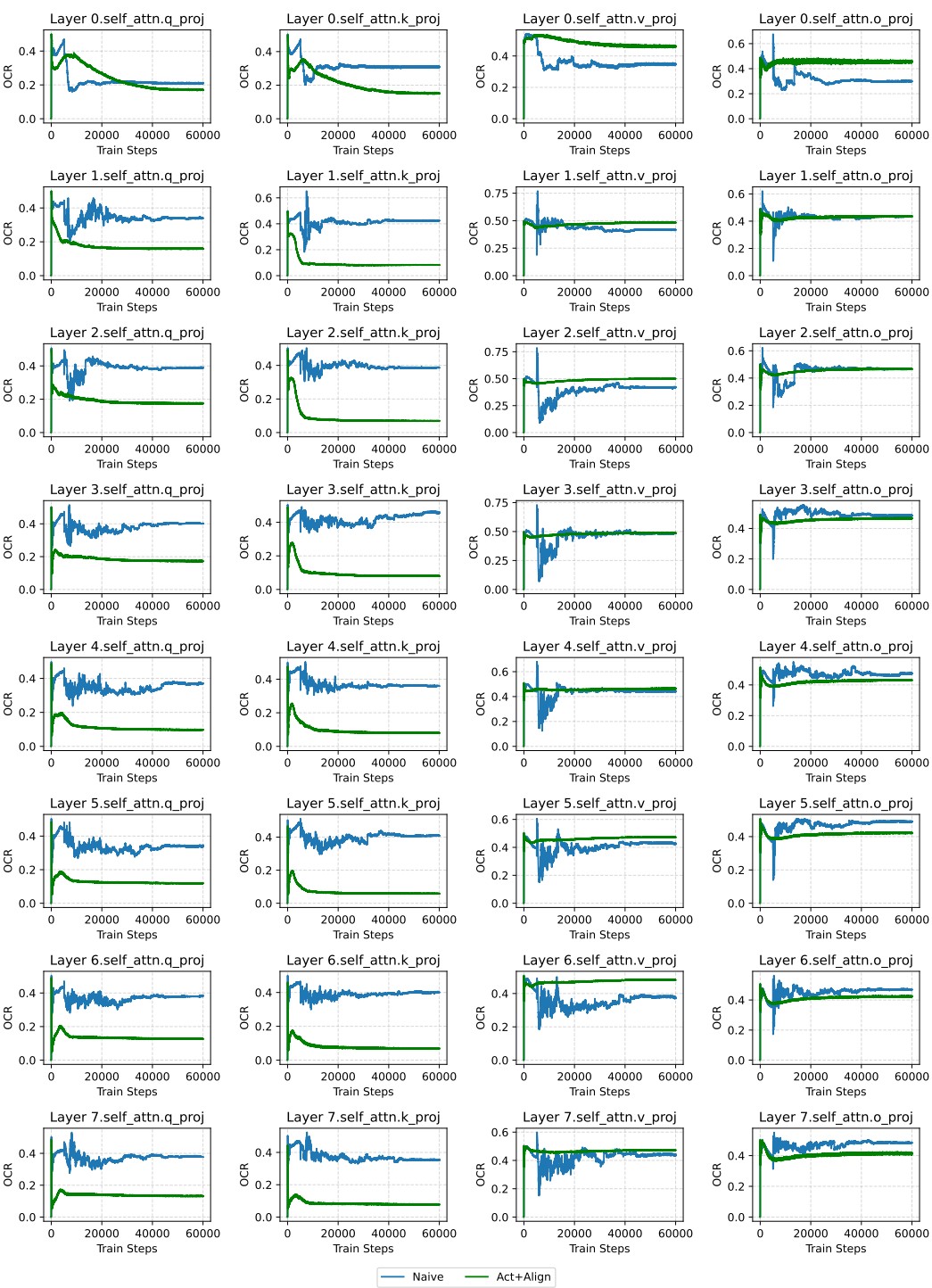

Figure 10: **The layer-wise output-level OCR plot** of LLaMA-350M on OpenWebText with a total sparsity of 0.9, with sparsity-configuration $s = 0.95, r = 32, d_{connectivity} : d_{spectral} = 1 : 1$. Each subplot in the figure reports the changes of OCR over training steps. For space limit, we report here the first 8 self-attention layers in the model, where each column refers to Q, K, V, O respectively.

