# OpenReview forum: "Alignment-Enhanced Integration of Connectivity and Spectral Sparsity in Dynamic Sparse Training of LLM"
_ICLR.cc/2026/Conference — ICLR 2026 Poster_

### Official Review · Reviewer_iVGQ · 2025-10-28

**Soundness:** 3
**Presentation:** 3
**Contribution:** 3
**Rating:** 6
**Confidence:** 4

**Summary:**

The paper explores a combination of LoRA, activation and dynamic sparse connections to improve pre-training efficiency. It attempts to unify CoLA and CHTs approaches through an alignment loss, aiming to maintain performance while reducing parameter usage.

**Strengths:**

1. The idea of integrating CoLA with dynamic sparse connections is conceptually reasonable and consistent with recent trends in parameter-efficient pre-training. The use of alignment loss to bridge different sparsity paradigms shows an effort toward unified optimization.

2. The experiments show solid performance across different sparsity levels, demonstrating the stability and robustness of the proposed approach.

3. The paper is well-written and clearly structured, allowing readers to easily follow the methodology and experimental design.

**Weaknesses:**

1. The proposed method mainly unifies CoLA and CHTs via an alignment loss, which appears to be an incremental combination and is somewhat limited in novelty.
2. The evaluation is somewhat limited. It would be beneficial to include downstream task benchmarks such as HellaSwag and COPA, similar to what was done in CHTS, to better demonstrate generalization capability and practical value.

**Questions:**

None.

---

> ### Author Response · Authors · 2025-11-21
> **Response to Reviewer iVGQ on Weakness 1 (Part 1)**
>
> ## **Reply to weakness1 (part 1)**
> *"The proposed method mainly unifies CoLA and CHTs via an alignment loss, which appears to be an incremental combination and is somewhat limited in novelty. "*
>
> We appreciate the reviewer’s comment and would like to clarify that our work goes beyond an incremental combination of existing ideas. Our contributions introduce several aspects that, to our knowledge, have not been explored in prior sparse training research.
>
> 1. **First integration of connectivity sparsity and spectral sparsity** within a dynamic sparse training framework. We make the first attempt to jointly incorporate dynamic connectivity sparsity and low-rank (spectral) sparsity, which differs from prior work such as SLTrain, where static connectivity sparsity played only a supplementary role. For empirical observation, CHTsL benefits a lot from the dynamic connectivity sparsity, **achieving higher performance compared with SLTrain under different settings**, as shown here which is from Table2 in our article, **with a 12/12 win-rate**.
> |Dataset|Method||LLaMA-60M|||LLaMA-130M||
> |-|-|-|-|-|-|-|-|
> |||s_total=0.9|s_total=0.8|s_total=0.7|s_total=0.9|s_total=0.8|s_total=0.7|
> |OpenWebText|SLTrain|33.90|29.83|27.86|25.33|22.81|21.25|
> ||CHTsL|**31.77**|**29.11**|**27.40**|**24.07**|**21.87**|**20.65**|
> |C4|SLTrain|41.05|37.00|34.89|31.38|28.28|26.78|
> ||CHTsL|**39.29**|**35.95**|**34.19**|**30.03**|**27.59**|**26.19**|
>
> 2. **Identification of cancellation and introducing OCR as a quantitative measure.** We observe that simply combining sparse and low-rank branches can lead to a **cancellation effect** that weakens model expressivity. To better understand this issue, we **introduce the overlap cancellation ratio (OCR)** as a quantitative measure. In the main text of article, Figure2, we sufficiently discussed about how OCR rate is changed with different integration strategies, with **Q,K layers’ behavior observed with significant change** in the cancellation effect when applying alignment training scheme.  Also, we conducted additional experiments to show that this observation based on OCR value is of practical value, where we compare alignment loss applied on Q/K, layers except Q/K layers, and on all layers. The new results can be found here and in Appendix G, we show that the **computational cost can be further improved** by only applying alignment loss on Q/K layers, with an similar performance compared with applying on all layers, **credited to the OCR observations**.
> |Model|Dataset|TotalSparsity|Align_qk|Align_other|Align_all|
> |-|-|-|-|-|-|
> |LLaMA-60M|OpenWebText|0.9|32.012|32.224|**31.772**|
> |||0.8|**29.066**|29.353|29.109|
> |||0.7|**27.279**|27.802|27.400|
> ||C4|0.9|39.376|39.713|**39.291**|
> |||0.8|**35.860**|36.281|35.949|
> |||0.7|34.194|34.437|**34.191**|
> |LLaMA-130M|OpenWebText|0.9|24.251|24.400|**24.071**|
> |||0.8|21.878|**21.772**|21.866|
> |||0.7|20.690|26.655|**20.648**|
> ||C4|0.9|30.135|30.454|**30.034**|
> |||0.8|**27.567**|27.798|27.593|
> |||0.7|**26.143**|38.250|26.190|
> |Average Score|||29.038|30.762|**29.010**|
> |Win Rate|||0.42|0.08|**0.5**|
> |signed-rank|against|Align_qk|\\|0.00098|0.62207|
> |p-value|against|Align_other|\\|\\|0.00098|
>
> 3. **Proposing the alignment loss for better cooperation between branches.** To effectively solve the problem of cancellation, we **propose an alignment-based integration** scheme that encourages cooperation between the branches. The alignment loss is designed specifically to alleviate the observed cancellation phenomenon rather than serving as a generic auxiliary term. The alignment training scheme is verified by CHTsL, with **the performance improved significantly (p-value < 0.05)** compared with the naive integration of CHTs and CoLA (Act), as shown in Table 1 of the main text. We **additionally tested the alignment scheme using the combination of “static sparse+low-rank”**, both here and in Appendix E. Results verified that the alignment scheme sufficiently improves the performance compared with simple integration, which **further demonstrates the generality of the scheme**, which is not an incremental work but a training scheme of practical value in different scenarios.
> |Model|Dataset|TotalSparsity|Naive|Act (CoLA+CHTs)|Act+Align(CHTsL)|
> |-|-|-|-|-|-|
> |LLaMA-130M|OpenWebText|0.9|31.52|25.44|**25.41**|
> |||0.8|22.44|22.37|**22.36**|
> |||0.7|21.25|41.51|**20.97**|
> ||C4|0.9|31.49|31.62|**31.44**|
> |||0.8|28.21|28.35|**28.12**|
> |||0.7|26.90|26.52|**26.36**|
> |Wilcoxon signed-rank|against|Naive|\\|1|0.03125|
> |p-value|against|Act|\\|\\|0.03125|

---

> ### Author Response · Authors · 2025-11-21
> **Response to Reviewer iVGQ on Weakness 1 (Part 2) and Weakness 2**
>
> ## **Reply to Weakness1 (Part 2)**
> *"The proposed method mainly unifies CoLA and CHTs via an alignment loss, which appears to be an incremental combination and is somewhat limited in novelty. "*
>
> 4. **Strong empirical performance.** Our instantiation, CHTsL, integrates advanced CHTs with low-rank factorization. Experiments across datasets and model scales show that **CHTsL consistently achieves leading performance among parameter-efficient approaches of comparable budgets**, approaching dense-model accuracy without introducing substantial overhead. Compared with the mentioned baseline CHTs and CoLA, **CHTs significantly outperforms them under different settings, with a win-rate 12/12** on LLaMA-60M and LLaMA-130M.  Additionally, according to our results on larger models (reported here and in Appendix H), CHTsL consistently achieves lower perplexity across different total sparsity levels compared to the baselines on LLaMA-350M. On LLaMA-1B, CHTsL outperforms all other methods at the 0.9 sparsity level, where the benefits of our approach become particularly pronounced, while at the 0.7 sparsity level it remains competitive. Overall, CHTsL demonstrates **robust performance on large-scale models**, underscoring its scalability and its strong advantage **especially under higher sparsity regimes**.
>
> |Dataset|Method||LLaMA-350M||LLaMA-1B||
> |-|-|-|-|-|-|-|
> |||s_total=0.9|s_total=0.8|s_total=0.7|s_total=0.9|s_total=0.7|
> |OpenWebText|Dense||14.90||14.62||
> ||CHTs|19.69|17.82|17.88|17.35|14.53|
> ||CoLA|20.92|17.60|16.13|16.03|**13.08**|
> ||SLTrain|18.99|16.88|15.98|16.00|14.58|
> ||CHTsL|**18.40**|**16.54**|**15.86**|**15.16**|13.31|
>
> We hope these clarifications help convey the conceptual motivation, practical value and novelty of our method.
>
> ## **Reply to Weakness2**
> *"The evaluation is somewhat limited. It would be beneficial to include downstream task benchmarks such as HellaSwag and COPA, similar to what was done in CHTS, to better demonstrate generalization capability and practical value."*
>
> We sincerely thank the reviewer for pointing this out. We agree that evaluating downstream tasks is important for demonstrating generalization capability and practical value. To address this, we conducted zero-shot evaluations on several downstream tasks using the models trained under a total sparsity of 0.9. We compared CHTsL with the strongest baselines, SLTrain and CHTs, and also included the dense model for reference. The results are summarized here and in Appendix J. They show that **CHTsL achieves the highest win rate, even surpassing the dense model** in these settings. We hope these results further address the reviewer’s concern.
> |Model|Pretrain|Downstream|CHTsL|SLTrain|CHTs|Dense|
> |-|-|-|-|-|-|-|
> |LLaMA-60M|OpenWebText|CoLA|0.5292|0.6894|**0.6913**|**0.6913**|
> |||Copa|0.5300|**0.5700**|**0.5700**|0.5200|
> |||Hellaswag|0.2653|0.2649|**0.2663**|0.2619|
> |||MNLI|0.3278|**0.3310**|0.3290|0.3282|
> |||MRPC|0.3235|0.3995|**0.6838**|0.6789|
> |||QNLI|**0.4955**|0.4935|0.4944|0.4946|
> |||QQP|**0.4126**|0.3688|0.3682|0.3682|
> |||RTE|**0.5235**|0.5126|0.4838|0.5018|
> |||SST-2|**0.5482**|0.4908|0.4908|0.4920|
> ||C4|CoLA|**0.6913**|0.6903|0.4276|0.6846|
> |||Copa|0.4800|0.5200|**0.5400**|0.4500|
> |||Hellaswag|0.2666|0.2644|**0.2714**|0.2656|
> |||MNLI|0.3255|**0.3340**|0.3291|0.3281|
> |||MRPC|0.6544|0.6495|0.6324|**0.6740**|
> |||QNLI|**0.4944**|0.4915|0.4926|0.4939|
> |||QQP|0.3682|0.3727|0.3692|**0.3730**|
> |||RTE|**0.5487**|0.4874|0.5271|0.5162|
> |||SST-2|0.4908|**0.4931**|0.4908|0.4908|
> |LLaMA-130M|OpenWebText|CoLA|0.6606|0.6568|**0.6913**|**0.6913**|
> |||Copa|0.5700|0.5600|0.5600|**0.6000**|
> |||Hellaswag|0.2687|0.2656|0.2678|**0.2699**|
> |||MNLI|0.3254|**0.3320**|0.3275|0.3272|
> |||MRPC|0.5662|0.3775|**0.6814**|0.6740|
> |||QNLI|**0.4961**|0.4950|0.4941|0.4946|
> |||QQP|**0.4015**|0.3762|0.3682|0.3717|
> |||RTE|**0.5018**|0.4477|0.4585|0.4765|
> |||SST-2|**0.5161**|0.4920|0.4908|0.4908|
> ||C4|CoLA|0.6002|0.6443|**0.6903**|**0.6903**|
> |||Copa|0.4700|0.4900|0.5300|**0.5700**|
> |||Hellaswag|0.2713|0.2694|0.2701|**0.2770**|
> |||MNLI|**0.3282**|0.3274|0.3276|0.3274|
> |||MRPC|**0.5686**|0.3211|0.4412|0.5074|
> |||QNLI|**0.5041**|0.5003|0.4946|0.4946|
> |||QQP|0.3720|**0.4326**|0.3683|0.3682|
> |||RTE|**0.5199**|0.5162|0.4946|0.5018|
> |||SST-2|0.4908|**0.4920**|0.4908|0.4908|
> |Win Rate|||**0.4167**|0.1944|0.25|0.25|
>
> In all, We sincerely appreciate the reviewer’s insightful comments regarding both the novelty of our method and the need for downstream evaluations. In response, we have provided a detailed clarification of the conceptual and methodological contributions that distinguish our approach from prior sparse training work, and we have conducted additional zero-shot downstream experiments to strengthen the empirical evidence. We hope that these explanations and new results satisfactorily address the reviewer’s concerns and demonstrate the originality and practical value of our method.

---

> > ### Comment · Reviewer_iVGQ · 2025-11-27
> >
> > Thank you for the detailed response, which has essentially addressed my concerns. The discussion from both performance comparison and fine-grained exploration perspectives has alleviated doubts regarding the novelty of this method. Moreover, experiments on downstream tasks were conducted, with average performance even surpassing that of dense models. I have raised my score to 8 to further endorse this paper. That said, I would like to ask if the authors could share some insights on why CHTsL outperforms dense models.

---

> > > ### Author Response · Authors · 2025-12-02
> > >
> > > We sincerely thank the reviewer for the positive feedback and for raising the score.
> > >
> > > Regarding the question of why CHTsL can outperform the dense model, one possible explanation is the **over-parameterization of dense models** especially those of large size. As shown in our results, at a sparsity level of 0.7 on LLaMA-1B, all sparse-training methods achieve better performance than the dense baseline. This phenomenon is consistent with previous literature where work like [1] also reported that sparse-training methods can surpass dense models.
> > >
> > > [1] Zhang Y, Cerretti D, Zhao J, et al. Brain network science modelling of sparse neural networks enables Transformers and LLMs to perform as fully connected[J]. arXiv preprint arXiv:2501.19107, 2025.

---

### Official Review · Reviewer_QMmU · 2025-10-31

**Soundness:** 2
**Presentation:** 2
**Contribution:** 2
**Rating:** 4
**Confidence:** 5

**Summary:**

This work proposed CHTSL, an approach that unifies connectivity sparse training and spectral sparse training. They introduces an alignment loss mitigate the disagreement between the two branches and promote better collaboration. Experiments on small-scale LLama validate the efficacy of their approach.

**Strengths:**

- The paper is written and organized well.
- The idea of alignment loss is reasonable and inspiring.

**Weaknesses:**

- Lack of literature review. The paper discussed with pruning works, accounting for connectivity sparse training, differentiating against spectral sparse (low-rank) training. But in my opinion, structured pruning actually results in low-rank for the full model. Since the structured pruned models would have many columns or rows zero-out. The paper needs to discuss with structured-pruning-aware works, such as Only-Train-Once.

Only Train Once: A One-Shot Neural Network Training And Pruning Framework

- Lack of sufficient numerical experiments.

  - The numerical results are conducted under small-scale LLMs. It would be better to conduct over larger-scale LLMs to show the generality.

  -  Besides SiLU, it would be better to show over other activations.

**Questions:**

See the weakness.

---

> ### Author Response · Authors · 2025-11-21
> **Response to Reviewer QMmU on Weakness**
>
> ## **Reply to Weakness1**
> *"Lack of literature review. The paper discussed with pruning works, accounting for connectivity sparse training, differentiating against spectral sparse (low-rank) training. But in my opinion, structured pruning actually results in low-rank for the full model. Since the structured pruned models would have many columns or rows zero-out. The paper needs to discuss with structured-pruning-aware works, such as Only-Train-Once."*
>
> We thank the reviewer for this useful comment, we now **revised the section 2.2** on low-rank spectral sparse training in the related work adding at the end of the section as follow:
> As a side note, while we previously discussed the relevance of pruning mainly in the context of connectivity-based sparse training, in contrast to spectral low-rank training, structured pruning can also be viewed as implicitly inducing a low-rank structure in the resulting model. This is because structured-pruned models remove entire channels or filters, which correspond to removing rows or columns in the unfolded weight matrices, thereby potentially reducing their effective rank. Representative structured-pruning-aware works include channel pruning via LASSO, network slimming, and **more recently Only-Train-Once (OTO)**, which explicitly consider structural constraints during training to improve efficiency for subsequent pruning.
>
> We hope that this addressed the reviewer’s concern of literature review.
>
> ## **Reply to Weakness 2.1**
> *"The numerical results are conducted under small-scale LLMs. It would be better to conduct over larger-scale LLMs to show the generality."*
>
> We sincerely thank the reviewer for the insightful question. We fully agree that evaluating the generalization of the alignment scheme to larger models is important. To address this concern, we conducted additional experiments with CHTsL on the LLaMA-350M and LLaMA-1B, using the OpenWebText dataset, comparing against the most competitive baselines, SLTrain, CHTs and CoLA, with dense model performance as a reference. The results are reported here and in Appendix H of the revised manuscript, where the detailed hyperparameter settings are also provided. The experiments show that **CHTsL consistently achieves lower perplexity** across different total sparsity levels compared to the baselines on LLaMA-350M. On LLaMA-1B, CHTsL outperforms all other methods at the 0.9 sparsity level, where the benefits of our approach become particularly pronounced, while at the 0.7 sparsity level it remains competitive. Overall, CHTsL demonstrates **robust performance on large-scale models**, underscoring its scalability and its strong advantage **especially under higher sparsity regimes**.
>
> |Dataset|Method||LLaMA-350M||LLaMA-1B||
> |-|-|-|-|-|-|-|
> |||s_total=0.9|s_total=0.8|s_total=0.7|s_total=0.9|s_total=0.7|
> |OpenWebText|Dense||14.90||14.62||
> ||CHTs|19.69|17.82|17.88|17.35|14.53|
> ||CoLA|20.92|17.60|16.13|16.03|**13.08**|
> ||SLTrain|18.99|16.88|15.98|16.00|14.58|
> ||CHTsL|**18.40**|**16.54**|**15.86**|**15.16**|13.31|
>
> ## **Reply to Weakness 2.2**
> *"Besides SiLU, it would be better to show over other activations."*
>
> We sincerely thank the reviewer for the insightful question. We applied the SiLU activation function in the low-rank branch for two main reasons:
> 1.for consistency with the LLaMA architecture, which uses SiLU as its activation function.
> 2.for fair comparison with the CoLA baseline, which also adopts SiLU in its low-rank training.
>
> We agree that discussing the effect of different activation functions is necessary. Therefore, we conducted additional ablation experiments on CHTsL by replacing SiLU with ReLU and GeLU in the low-rank branch. For fairness, all experiments were performed under the same hyperparameter settings as the original CHTsL (SiLU) reported in the paper. The results are provided here and in Appendix I of the revised manuscript. They show that **CHTsL with SiLU achieves the best performance in most cases**.
>
> |Model|Dataset|Total Sparsity|ReLU|GeLU|SiLU|
> |-|-|-|-|-|-|
> |LLaMA-60M|OpenWebText|0.9|32.431|32.081|**31.772**|
> |||0.8|29.607|29.280|**29.109**|
> |||0.7|28.125|35.625|**27.400**|
> ||C4|0.9|39.930|39.364|**39.291**|
> |||0.8|36.871|36.045|**35.949**|
> |||0.7|35.251|**34.157**|34.191|
> |LLaMA-130M|OpenWebText|0.9|24.530|**24.049**|24.071|
> |||0.8|22.219|21.999|**21.866**|
> |||0.7|20.999|20.789|**20.648**|
> ||C4|0.9|30.639|30.171|**30.034**|
> |||0.8|28.252|27.762|**27.593**|
> |||0.7|27.044|26.192|**26.190**|
>
> In all, we sincerely thank the reviewer for the insightful and constructive comments. We have made substantial efforts to revise the manuscript and conduct additional analyses, and we hope these updates have fully resolved the reviewer’s concerns.

---

### Official Review · Reviewer_ob7g · 2025-10-31

**Soundness:** 3
**Presentation:** 3
**Contribution:** 3
**Rating:** 6
**Confidence:** 4

**Summary:**

This paper studies the cancellation effect between connectivity-sparse and spectral-sparse branches in dynamic sparse training and introduces an alignment loss. The idea is simple and practical, and experiments show clear gains over baselines.

**Strengths:**

- This paper identifies the "cancellation effect" and proposes the OCR metric, which provides a valuable quantitative perspective on hybrid sparse training.

- The proposed method is simple and easy to implement.

**Weaknesses:**

- The alignment loss is conceptually orthogonal to any combination of dynamic sparsity and low-rank training, yet experiments are limited to the CHTs + low-rank setup. Testing additional combinations would be necessary to confirm its generality.

- OCR captures output-level discrepancies but does not fully demonstrate whether alignment mitigates gradient-level conflicts between branches. A more comprehensive analysis at the gradient level is recommended.

- The paper lacks practical efficiency evaluations such as inference memory and throughput.

**Questions:**

- Is there a correlation between OCR and global cosine similarity? Since OCR measures element-wise sign inconsistency, it may be influenced by local fluctuations rather than true directional cancellation. Such analysis could clarify OCR’s distinct role.

- How does the model's performance differ when the alignment loss is applied only to the Q/K layers compared to applying it across Q/K/V/O or FFN layers?

- Do larger models (e.g., LLaMA-7B) exhibit similar cancellation patterns, and does alignment maintain its effectiveness at scale?

---

> ### Author Response · Authors · 2025-11-21
> **Response to Reviewer ob7g on Weakness 1-3**
>
> ## **Reply to Weakness 1**
> *"The alignment loss is conceptually orthogonal to any combination of dynamic sparsity and low-rank training, yet experiments are limited to the CHTs + low-rank setup. Testing additional combinations would be necessary to confirm its generality."*
>
> Thank you to the reviewer for this helpful suggestion. We originally presented only the “CHTs + low-rank” combination because CHTs demonstrates clear advantages over other dynamic connectivity sparse training methods as shown in Table 2. Our goal was to validate the effectiveness of the alignment scheme on the strongest connectivity-sparse baseline.
>
> We fully agree that testing additional combinations would further validate the generality of the proposed alignment loss. To address this concern, we additionally evaluated the combination of “static sparse + low-rank”, both with and without the alignment scheme. The results are provided here and in Appendix E of the updated revision. All experiments are based on LLaMA-130M with a sparsity configuration of $d_{\text{connectivity}} : d_{\text{spectral}} = 1:1$ and $\lambda = 0.3$. The results show that **alignment consistently improves performance** compared with the Naive combination of two branches, with the Wilcoxon signed-rank test yielding a p-value below 0.05.
>
> |Model|Dataset|Sparsity|Naive|Act|Act+Align|
> |-|-|-|-|-|-|
> |LLaMA-130M|OpenWebText|0.9|31.52|25.44|**25.41**|
> |||0.8|22.44|22.37|**22.36**|
> |||0.7|21.25|41.51|**20.97**|
> ||C4|0.9|31.49|31.62|**31.44**|
> |||0.8|28.21|28.35|**28.12**|
> |||0.7|26.90|26.52|**26.36**|
> |Wilcoxon signed-rank|against|Naive|\\|1|0.03125|
> |p-value|against|Act|\\|\\|0.03125|
>
> Also, we’re conducting more experiments to address the reviewer’s concern with other dynamic sparse training methods considered. As shown here and also in revised Appendix M, **alignment-enhanced training scheme effectively improves the performance compared with naive integration, using different connectivity sparse training methods** including static sparse training, SET and CHTs, with different initialization strategy.
>
> |Connectivity Sparsity|Initialization|Naive|Act+Align|
> |-|-|-|-|
> |Static|random|22.44|**22.36**|
> ||BSW|380.88|**21.88**|
> |SET|random|356.08|**22.54**|
> ||BSW|22.55|**22.22**|
> |CHTs|random|22.46|**22.20**|
> ||BSW|22.11|**21.87**|
>
>
> ## **Reply to Weakness2**
> *"OCR captures output-level discrepancies but does not fully demonstrate whether alignment mitigates gradient-level conflicts between branches. A more comprehensive analysis at the gradient level is recommended."*
>
> As suggested by the reviewer, we explored how gradients behave under the alignment loss. Among the possible approaches, we focused on the most feasible one: examining the gradients with respect to the branch inputs (i.e., the gradients of x with respect to two branches). We report in Appendix L the overlap cancellation ratio (OCR) between these input gradients. The results show that, at the gradient level, certain layers exhibit a reduced cancellation effect (e.g., the 5th Q layer). However, **the phenomenon is less pronounced than at the output level**, which is expected since **the alignment loss directly acts on the outputs rather than on the gradients**.
>
> ## **Reply to Weakness3**
>
> *"The paper lacks practical efficiency evaluations such as inference memory and throughput."*
>
> Thank you for the reviewer’s suggestion. For inference, we used the SLTrain C++ codebase, which provides optimized sparse and low-rank kernels. For CHTsL, we applied two minimal modifications: adding the activation function and correcting the low-rank computation order in SLTrain (the original implementation computes B@A and then B@A@X, introducing redundant operations).
>
> As reported in Appendix K and here, **CHTsL achieves both lowest memory usage and highest throughput**. This shows that CHTsL already offers practical efficiency gains while **retaining even greater theoretical potential**, since it holds even greater promise as software and hardware support for unstructured sparsity improves.
>
> |Model|Method|Memory(GB)|Throughput(Tokens/s)|
> |-|-|-|-|
> |LLaMA-60M|Dense|2.606|773111|
> ||SLTrain|**2.573**|749985|
> ||CHTsL|**2.573**|**786004**|
> |LLaMA-130M|Dense|3.392|343310|
> ||SLTrain|**3.278**|337149|
> ||CHTsL|**3.278**|**386483**|
>
> We hope this clarification addresses the reviewer’s concern.

---

> ### Author Response · Authors · 2025-11-21
> **Response to Reviewer ob7g on Question 1-2**
>
> ## **Reply to Question1**
> *"Is there a correlation between OCR and global cosine similarity? Since OCR measures element-wise sign inconsistency, it may be influenced by local fluctuations rather than true directional cancellation. Such analysis could clarify OCR’s distinct role."*
>
> Thank you for the reviewer’s question. OCR is fundamentally different from global cosine similarity. Cosine similarity measures only directional alignment and inherently ignores magnitude, which is crucial for reflecting the severity of cancellation. In contrast, OCR directly quantifies how much signal is actually lost in overlapping regions.
>
> This distinction is illustrated by the following two examples, **both with cosine similarity = 0**:
> - Example A (strong cancellation): S=[10,−10], L=[10,10]. S+L=[20,0] shows half cancellation and **OCR around 0.5**.
> - Example B (almost no cancellation): S=[100,0], L=[0,100]. S+L=[100,100] shows no cancellation and **OCR = 0**.
>
> These examples demonstrate that cosine similarity alone cannot capture the magnitude of cancellation, while OCR effectively quantifies it, highlighting the necessity of OCR in measuring cancellation severity.
>
> To stress the distinct role of OCR in identifying the severity of cancellation, we **add the difference between cosine similarity and OCR** in Appendix F of the revision, which would make it clear to the readers. Additionally, we agree that it would be a good supplement to reflect the global-directional-alignment using cosine similarity, and we **include the cosine similarity plot** in Appendix F of revision. As shown in the plot, alignment loss truly improves the alignment in direction for the output of different branches especially on Q,K layers with significantly higher cosine similarity.
>
> ## **Reply to Question2**
> *"How does the model's performance differ when the alignment loss is applied only to the Q/K layers compared to applying it across Q/K/V/O or FFN layers?"*
>
> We sincerely thank the reviewer for this insightful question. During our experiments, we observed that the alignment loss significantly affects the OCR in the Q/K layers. Following the reviewer’s suggestion, we conducted additional experiments applying the alignment loss only to the Q/K layers (Align_qk) and only except Q/K layers (Align_others) with respect to the original results (Align_all) , using the same hyperparameter settings as in the original experiments for a fair comparison. The results, reported here and in Appendix G of the revised manuscript, indicate that **restricting the alignment loss to the Q/K layers yields similar performance**. **Align Q/K layers significantly outperforms align other linear layers**, with the Wilcoxon signed-rank p-value lower than 0.05. This finding is particularly encouraging, as it reduces computational cost while highlighting the critical role of alignment in the Q/K layers, which also **proves the effectiveness of OCR in reflecting the cancellation**.
>
> |Model|Dataset|TotalSparsity|Align_qk|Align_other|Align_all|
> |-|-|-|-|-|-|
> |LLaMA-60M|OpenWebText|0.9|32.012|32.224|**31.772**|
> |||0.8|**29.066**|29.353|29.109|
> |||0.7|**27.279**|27.802|27.400|
> ||C4|0.9|39.376|39.713|**39.291**|
> |||0.8|**35.860**|36.281|35.949|
> |||0.7|34.194|34.437|**34.191**|
> |LLaMA-130M|OpenWebText|0.9|24.251|24.400|**24.071**|
> |||0.8|21.878|**21.772**|21.866|
> |||0.7|20.690|26.655|**20.648**|
> ||C4|0.9|30.135|30.454|**30.034**|
> |||0.8|**27.567**|27.798|27.593|
> |||0.7|**26.143**|38.250|26.190|
> |Average Score|||29.038|30.762|**29.010**|
> |Win Rate|||0.42|0.08|**0.5**|
> |signed-rank|against|Align_qk|\\|0.00098|0.62207|
> |p-value|against|Align_other|\\|\\|0.00098|
>
> We sincerely thanks the reviewer for this insightful suggestion which further strengthens this work in an elegant way.

---

> ### Author Response · Authors · 2025-11-21
> **Response to Reviewer ob7g on Question 3**
>
> ## **Reply to Question3**
> *"Do larger models (e.g., LLaMA-7B) exhibit similar cancellation patterns, and does alignment maintain its effectiveness at scale?"*
>
> We sincerely thank the reviewer for the insightful question. We fully agree that evaluating the generalization of the alignment scheme to larger models is important. To address this concern, we conducted additional experiments with CHTsL on the LLaMA-350M and LLaMA-1B, using the OpenWebText dataset, comparing against the most competitive baselines, SLTrain, CHTs and CoLA, with dense model performance as a reference. The results are reported here and in Appendix H of the revised manuscript, where the detailed hyperparameter settings are also provided. The experiments show that **CHTsL consistently achieves lower perplexity** across different total sparsity levels compared to the baselines on LLaMA-350M. On LLaMA-1B, CHTsL outperforms all other methods at the 0.9 sparsity level, where the benefits of our approach become particularly pronounced,  while at the 0.7 sparsity level it remains competitive. Overall, CHTsL demonstrates **robust performance on large-scale models**, underscoring its scalability and its strong advantage **especially under higher sparsity regimes**.
>
> |Dataset|Method||LLaMA-350M||LLaMA-1B||
> |-|-|-|-|-|-|-|
> |||s_total=0.9|s_total=0.8|s_total=0.7|s_total=0.9|s_total=0.7|
> |OpenWebText|Dense||14.90||14.62||
> ||CHTs|19.69|17.82|17.88|17.35|14.53|
> ||CoLA|20.92|17.60|16.13|16.03|**13.08**|
> ||SLTrain|18.99|16.88|15.98|16.00|14.58|
> ||CHTsL|**18.40**|**16.54**|**15.86**|**15.16**|13.31|
>
> Also, to test how alignment loss affects cancellation between branches when applied to larger model, we conducted additional experiments on LLaMA-350M, including the naive integration which simply integrates two branches and alignment-enhanced integration which utilizes activation and alignment loss. The corresponding results and output-level OCR plot can be found in Appendix N. Results show that alignment scheme effectively mitigates the cancellation effect, with better performance and dropped overlap cancellation ratio (OCR). These findings are **consistent with those observed on the smaller model**, further confirming the observation on cancellation effects at scale.
>
> In all, we sincerely thank the reviewer for the valuable suggestions. The comments were truly insightful and have helped us improve and strengthen this work. We hope that our revisions and additional analyses adequately address the reviewer’s concerns.

---

### Author Response · Authors · 2025-12-03
**Summary for Discussion**

# Summary for Discussion
We sincerely thank all reviewers for their valuable time and insightful feedback, which is helpful in further strengthening this work.
We are grateful that the reviewers appreciate:
1. The proposed method CHTsL shows “***solid performance***” and “***clear gains***” “***across different sparsity levels, demonstrating the stability and robustness***”. (ob7g, iVGQ)
2. The identified cancellation effect and proposed OCR metric, provide “***valuable quantitative perspective on hybrid sparse training***”. (ob7g)
3. Proposed alignment loss which “bridges different sparsity paradigms” is “***reasonable***”, “***inspiring***”, “***showing an effort towards unified optimization***”. (QMmU, iVGQ)
4. The paper is “***well-written and clearly structured***”. (QMmU, iVGQ)

To provide greater clarity on the revisions made to our paper and the experiments we conducted to address the reviewers’ questions, we have summarized the modifications and experiments made during the rebuttal period as follows:

**Additional Experiments:**
1. *Scaling to larger models*. CHTsL consistently outperforms baselines on LLaMA-350M, achieving to be the best (at 90% sparsity) or second best (comparable to first, at 70% sparsity) on LLaMA-1B, demonstrating **strong generality at scale especially under high sparsity level**.
2. *Downstream evaluation*. Across diverse downstream tasks, CHTsL achieves the **highest win rate (0.4167), surpassing even the dense model (0.25)**, further confirming its **effectiveness and generalization ability**.
3. *Inference memory and throughput test*. CHTsL exhibits **less peak memory usage** and achieves **higher throughput** than the dense model, demonstrating its advantages in **memory efficiency and practical inference efficiency**.
4. *Applying alignment loss on different combinations* of dynamic sparse training methods and low-rank training. Alignment enhanced training consistently improves the performance over naive integration, with **signed-rank p-value < 0.05**, demonstrating the **broad applicability** of the alignment-loss design **in combining different sparse training paradigms**.
5. *Ablation test on applying alignment loss to different layers*. We observed by OCR analysis that Q,K layers exhibit the most pronounced reduction in cancellation, motivating a targeted design. We further investigate **applying alignment loss solely to Q,K layers**, which achieves performance comparable to applying it to all layers, **offering insights** for more efficient alignment implementation strategies in future study.
6. *Ablation test on the choice of activation function* in the low-rank branch. It shows that SiLU performs the best, which is the one utilized in the article and is consistent with LLaMA structure.

**Clarifications:**
1. We add the discussion of structured-pruning methods as a side note in the related work section, clarifying the difference between pruning and sparse training.
2. We add a detailed discussion in Appendix F of **how the proposed OCR metric is important** in identifying the cancellation effect **by examples**, which clarifies that OCR is of fundamental difference compared with global cosine similarity in capturing the severity of cancellation between branches, because it accounts also for magnitude.
3. We discussed in Appendix L how the gradient-level OCR changes, which is of less obvious signs since the alignment-loss works directly on the output-level.
4. We **strengthen our claim of the novelty of CHTsL** from both performance comparison and fine-grained exploration perspectives in our response to reviewer iVGQ.

Finally, we want to highlight that we are dedicated to solving each comment of every reviewer, making sure that **each comment is with a detailed response**. Our effort in improving this work **has got the acknowledgement before reverting (for instance, reviewer iVGQ raised the score from 6 to 8)**. We believe that changes and responses we made would sufficiently address all the concerns and suggestions raised by reviewers, thereby enriching the quality of this work.

---

### Meta-Review · Area_Chair_XvaF · 2025-12-21

**Summary:**

The reviewers broadly agree the paper presents a practical hybrid sparse-training recipe that combines connectivity sparsity with spectral/low-rank sparsity, and that the “cancellation effect” framing plus the proposed metric (OCR) is a useful way to reason about failures in naïvely combining sparsity paradigms.  Overall, the rebuttal meaningfully strengthened the empirical story (scale, downstream tasks, efficiency, broader combinations) and improved positioning. I thus recommend acceptance.

**Reviewer Concerns:**

Concerns on limited evaluation, scalability, practical efficiency have been addressed during rebuttal. Concerns on deeper insights on why the method works and gradient-level conflicts are still valid.

**Reviewer Scores:**

One reviewer has raised the score from 6 to 8. And the other two reviewers may also increase the ratings given the major concerns are addressed in the rebuttal.

---

### Decision · Program_Chairs · 2026-01-26

Accept (Poster)